# Planning with an Embodied Learnable Memory

Priyam Parashar[*]   Jacob Krantz[*]   Matthew Chang[*]   Kavit Shah
Xavier Puig[†]   Roozbeh Mottaghi[†]

## Abstract

We develop a novel memory representation for embodied planning models performing long-horizon mobile manipulation in dynamic, large-scale indoor environments. Prior memory representations fall short in this setting, as they struggle with object movements, suffer from computational deficiencies, and often depend on the heuristic integration of multiple models. To overcome these limitations, we present the Embodied Perception Memory (EPM), a learnable memory designed for embodied planning. EPM is implemented as a unified Vision-Language Model (VLM) that uses egocentric vision to maintain and update a textual environment representation. We further introduce two complementary methods for training planners to leverage the EPM: an imitation strategy that uses human trajectories for natural exploration and interaction, and a novel reinforcement learning approach, Dynamic Difficulty-Aware Fine-Tuning (DDAFT), which improves planning performance via difficulty-aware exploration. Our memory representation, when integrated with our planning training methods, leads to significant improvements on planning tasks, showing up to a 55% increase in success rate on the PARTNR benchmark compared to strong baselines. Also, our planning method outperforms these baselines even when they have access to groundtruth perception.

## 1 Introduction

Imagine a human doing tasks in their home. Without even thinking, they can move towards the cabinet where they last placed a clean mug and plate, bring them to the table, and prepare a coffee to start the day. Our ability to perform these tasks effortlessly rely on an interplay between memory, perception, and planning. Memory provides us with a mental map of the objects in the environment, which we update based on our observations, and use to plan effectively. For robots to perform tasks in indoor environments with such efficiency, we need to equip them with the same capabilities: perceive surroundings, remember where objects are over long periods of time, and plan actions accordingly.

Developing an effective memory representation for embodied planning models that tackle long-horizon tasks poses significant challenges. The memory must not only capture the spatial configuration, context and state of objects in the scene, but also update dynamically as agents explore or the environment changes. Further, the memory must integrate with a planning system to provide context that enables the agent to perform a task.

The current literature on memory representations for planning in embodied tasks has certain limitations. A primary shortcoming of many existing approaches is their inability to effectively handle dynamic environments (Gu et al., 2024a; Jatavallabhula et al., 2023), where objects are frequently moved and changed by the robot or humans in the environment. Many of the existing approaches encode the environment by storing the agent's egocentric observations (Liu et al., 2024a; Yang et al., 2025), or a featurized point-cloud of the scene (Jatavallabhula et al., 2023; Liu et al., 2024a), which can be queried to retrieve relevant information. While multi-stage pipelines combining standalone detection and segmentation models with planners have been effective, particularly in complex scenes with small or rare objects, such representations often have high memory requirements and computational cost when querying large models multiple times for planning. Furthermore, formulating precise queries can be challenging and impractical for the planner. As found in prior work (Chang et al., 2024), naive feature matching with language queries fails to achieve both high precision and

---

[*] [†] Equal contribution. Work done at FAIR, Meta.

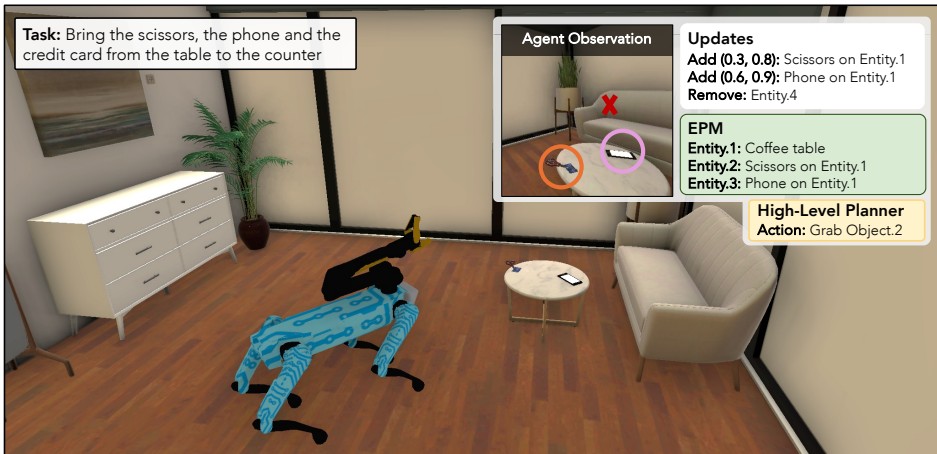

Figure 1: We present a memory representation for embodied planing in large-scale environments. Our method includes Embodied Perception Memory (EPM), a VLM-based system that represents the environment from egocentric observations. Our novel LLM-based planner trained with human demonstrations and RL reason over EPM to plan agent actions.

recall. To address these challenges, we propose a novel memory representation that is adaptable, efficient, can handle dynamic environments, and easily interacts with task planners.

In particular, we introduce the Embodied Perception Memory (EPM), a novel form of learnable external memory for planning in mobile manipulation tasks. EPM is a single Vision-Language Model (VLM) trained to perform various operations—such as adding, removing, and updating information—on scene representations derived from egocentric observations. This end-to-end approach eliminates the need for heuristics and enables faster inference than approaches consisting of multiple models. EPM encodes an object-centric representation of the environment, where each object is characterized by 3D coordinates and a natural language description of the object's state and context. This naturally results in a textual representation of the scene, which can be used by LLM-based planners that have demonstrated strong performance in planning tasks.

This representation enables the planner to access environment information directly, without the need for explicit, task-based queries (in contrast to approaches that rely on such queries, e.g., Yang et al. (2025)). Hence, the planner can focus only on generating actions to interact with the environment, rather than issuing separate API calls to retrieve information. As a result, the planner can be improved with robot interaction data where such queries may not be available. To leverage this, we introduce a strategy to train the planning model on human demonstrations of mobile manipulation tasks, resulting in improved exploration and task execution. Then, we develop a novel online RL-based approach that leverages insights from LLM reasoning works (Shao et al., 2024; Yuan et al., 2023; Tong et al., 2024) to improve planning from online experience. Specifically, we propose Dynamic Difficulty-Aware Fine-Tuning (*DDAFT*), which is a novel form of value-function-free RL for LLMs that explores the planning space on difficult instructions with online sampling. This induces a curriculum over the training data, leading to sample efficient policy improvement.

We evaluate EPM's ability to track the environment in simulated and real-world egocentric interaction datasets, showing that our approach can track the environment in both domains despite being trained only in simulation. We also evaluate EPM together with our proposed planner in PARTNR (Chang et al., 2024), a simulation benchmark measuring agents' abilities to perform tasks specified by language. Our results show significant improvements over strong baselines such as Liu et al. (2024a) and Chang et al. (2025), achieving absolute success rate gains of 55% and 12%, respectively, on planning tasks. Additionally, we show that human demonstration training outperforms zero-shot baselines even when zero-shot planners are provided groundtruth perception. Further, our DDAFT approach improves performance across all settings, providing a recipe to enhance embodied planning. In summary, our contributions are:

- We introduce the Embodied Perception Memory, a novel form of learnable memory for embodied planning tasks that effectively handles dynamic environments as a single model, and can be seamlessly integrated with LLM-based planners.

| Method | Open Vocabulary | Object Relationships | Error Correction | Dynamics | No Query | Single Model |
|---|---|---|---|---|---|---|
| ConceptFusion (Jatavallabhula et al., 2023) | ✓ | | | | | |
| ConceptGraphs (Gu et al., 2024a) | ✓ | ✓ | | | ✓ | |
| 3D-Mem (Yang et al., 2025) | ✓ | ✓ | ✓ | | | |
| DynaMem (Liu et al., 2024a) | ✓ | ✓ | ✓ | ✓ | | |
| EPM (ours) | ✓ | ✓ | ✓ | ✓ | ✓ | ✓ |

Table 1: **Comparison to different memory representations**. We compare EPM with other popular embodied memory representations. **Object Relationships:** whether object-furniture relationships are encoded. **Dynamics:** whether the representation allows for a changing environment. **No Query:** whether the representation needs to be queried to retrieve information.

- We present a novel planning approach that leverages human demonstration data and *DDAFT*, demonstrating significant performance improvements over strong baselines.

## 2 RELATED WORK

**Embodied Planning with LLMs and VLMs.** A large body of work has studied LLMs as policies for embodied agents, leveraging their strong capabilities in planning and common sense reasoning, but also highlighting limitations in grounding them to an agent's observations and actions (Ahn et al., 2022; Huang et al., 2023c; Zeng et al., 2022). To improve grounding, some methods integrate language models with external modules, which are exposed to the LLMs as APIs to reason over the agent's observations and commands (Schick et al., 2023). These modules can consist of learned or hand-designed low-level controllers (Wang et al., 2024; Driess et al., 2023), or perception models (Liu et al., 2024a; Chang et al., 2025; Rana et al., 2023). DynaMem (Liu et al., 2024a) maintains a point-cloud representation of the environment, which is queried by an LLM to obtain task-relevant information. As a result, planning with such representations requires an LLM to generate both text queries to extract information from the environment and high-level actions. Other approaches build a text representation of the environment in the form of a scene graph, which is used by the LLM for planning (Gu et al., 2024a; Rana et al., 2023; Chang et al., 2025). This separation decouples planning from perception and interaction, but is sensitive to the representations of each module, and errors which propagate throughout the system, resulting in planning failures (Chang et al., 2025; Li et al., 2025). Additionally, maintaining multiple specialized modules introduces computational overhead that increases overall system requirements. Furthermore, they struggle to model dynamic scenes, requiring heuristic to track objects over time. More recently, multiple works have used VLMs to predict agent actions directly from observations, either by predicting parameters for pre-defined robot skills (Nasiriany et al., 2024; Huang et al., 2023b; 2024) or finetuning them to directly predict low-level actions (Shi et al., 2025; Bjorck et al., 2025). These approaches ground actions directly into visual observations, but are limited by VLMs' memory constraints, and thus insufficient for tasks requiring long-horizon exploration in large environments. Our work is closer to the first family of approaches, but instead of relying on off-the-shelf models, we propose learnable a memory system trained on embodied data, and propose an approach to allow LLM planners to effectively work with noisy memory representations.

**Spatio-Semantic Memory.** There is a long history of research on spatio-semantic memory, i.e. remembering the locations of elements of the scene jointly with semantic information about those elements. Early works (Aydemir et al., 2013; Gupta et al., 2017) focus primarily on navigation and are not designed to handle open vocabulary instructions. A common line of work uses a VLM such as CLIP (Radford et al., 2021) or LSeg (Li et al., 2022) to produce language-aligned features for views of the scene. These features are typically projected into 3D using a point cloud (Huang et al., 2023a; Ha & Song, 2022; Liu et al., 2024a; Jatavallabhula et al., 2023), neural radiance field (Kerr et al., 2023; Qiu et al., 2024), or Gaussian splat (Ji et al., 2024; Shorinwa et al., 2024). While these methods permit open-vocabulary queries over the scene, such features do not natively encode inter-entity relationships making referential or contextual queries harder to address. To interface with a planning system, these approaches rely on matching spatial feature vectors with some query (typically via cosine similarity), querying a VLM on stored images (Yang et al., 2025), or store per-object features or labels based on location information (Gu et al., 2024a; Maggio et al., 2024; Chang et al., 2024). In contrast, EPM directly produces text representing entities and entity-relationships, allowing seamless

integration with an LLM planner. Further, all of these works struggle with object re-association and correction of spurious objects, relying on heuristics or thresholds. Re-association and correction are learned within EPM, avoiding these issues. Short-horizon planning over these representations may be achieved by transforming the current scene to a goal image (Wang et al., 2023), or leveraging a differentiable scene representation (Bolte et al., 2023). However, long-horizon planning with contextual features is most commonly performed with a pre-trained LLM. This can be done by finding locations via cosine similarity with queries produced from the LLM (Chen et al., 2023; Yan et al., 2025), or explicitly building a text representation of the scene (Gu et al., 2024a; Koch et al., 2024). EPM uses the latter approach, using the descriptions produced by the learned model directly in the context of an LLM. In Table 1 we provide an overview of different memory representations.

Finally, several works build dynamic representations during simultaneous localization and mapping (SLAM) (Jiang et al., 2024; Wu et al., 2024; Fan et al., 2022; Yu et al., 2018). Such representations generally fail to meet the needs of embodied planning: e.g., object re-initialization instead of tracking, limited vocabularies, and insufficient interpretability. We expand this discussion in Appendix A.1.

# 3 THE EMBODIED PERCEPTION MEMORY (EPM)

We describe the memory component of our system which builds an environment representation from ego-centric observations. We define the setting under which the memory operates and introduce the EPM to build and maintain a dynamic representation of the environment.

## 3.1 PROBLEM SETTING

We aim to detect and track objects in the scene as the agent interacts with the environment. Formally, let $M^t$ denote the environment representation at time $t$. We learn an update function $f$ such that $M^t = f(M^{t-1}, o^t, a^t)$, where $o^t, a^t$ are observations and actions. Dynamic objects are unknown and can be moved as the agent executes the task. Below, we define the environment state, observations, and actions.

**Environment State.** We represent the environment as a list of entities corresponding to the furniture and objects. Entity at index $i$ is associated with a unique $id_i$, a 3D coordinate $c_i \in \mathbb{R}^3$ of the approximate centroid of the object or furniture, and a natural language description $d_i$ containing information such as open-vocabulary name, state or relationship with other entities.

**Observations.** At every timestep $t$, the agent is given egocentric observations $o^t$ from the environment: RGBD images, the camera pose, and intrinsic parameters.

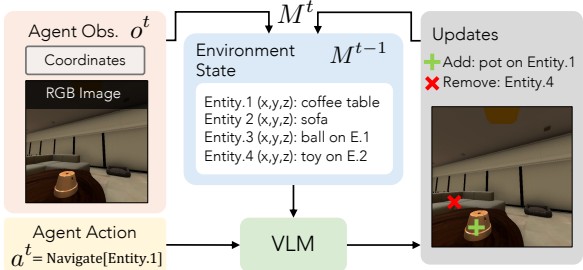

Figure 2: Overview of the EPM.

**Actions.** The agent takes high-level actions in the environment. Action $a$ is a tuple $(u, id_{v_1} \ldots id_{v_n})$ where $u$ is a verb (e.g., Navigate, Pick, Place) and $v_1 \ldots v_n$ index into the entities of environment state $M$. We use the same action space as Chang et al. (2025) (Details in Appendix A.2 and A.4).

## 3.2 TRACKING THE ENVIRONMENT STATE WITH THE EPM

Instead of generating a new environment state $M^t$, the EPM outputs a set of discrete operations that are combined with $M^{t-1}$ and the agent's egocentric observations to obtain $M^t$. We consider four operation types (we first list the format of the operation, then describe the function):

- `Add (<coords>):<description>`: Adds a new entity to $M^{t-1}$, with the natural language description $d = $ `<description>`. The entity coordinates $c$ are derived by unprojecting the pixel coordinates `<coords>` into the world frame.
- `Update k (<coords>):<description>`: Updates the entity with index $j$ in $M^{t-1}$ such that $id_j = k$, modifying its coordinates and description.

- `Remove k`: Removes the entity at index $j$ from $M^{t-1}$ such that $id_j = k$.
- `No updates`: Maintains the same memory state $M^t \leftarrow M^{t-1}$.

The `Add` operation appends previously-unseen objects to the environment state. `Update` tracks changes of a given object (for instance, if a cabinet was opened or an object was grabbed), and enables correction of perception errors (e.g., correcting the category of a mislabeled object once it is seen from a closer distance). `Remove` removes entities that have disappeared from the scene or were previously misdetected. If there is no new information, the EPM outputs `No updates`. We train a VLM to predict EPM updates. Given the RGB observation, textual environment state, robot action, and agent's pose, the model predicts a set of the above operations that update the memory.

**Training**. We generate a dataset of agent interactions on PARTNR (Chang et al., 2025), a simulation benchmark requiring agents to explore and move objects in indoor environments. We initialize $M^0$ with the static environment layout and furniture (e.g., cabinets and large tables). We obtain this information directly from the simulator. Alternatively, we could leverage recent powerful techniques to obtain geometric layouts using structure-from-motion techniques (Wang et al., 2025; Tang et al., 2025), and furniture by systems that lift 3D reconstructions to static object instances, e.g. EgoLifter (Gu et al., 2024b). We initialize EPM with $M^0$ and use heuristics to derive a sequence of operations that update it over time using privileged information. This produces a dataset of observations, actions, environment states, and update operations. Using this as supervision, we finetune a `LLaVa-OneVision-7b` base model (Li et al., 2024) with LoRA (Hu et al., 2022).

## 4 Planning with an Embodied Memory

We develop a planner that leverages EPM memory to perform mobile manipulation tasks specified in natural language (e.g., tasks in Figure 1). We present the agent, its use of EPM, and two methods for training the planner to achieve increased task performance and perception robustness.

### 4.1 Architecture

We adopt a two-layer control architecture following Chang et al. (2025) as illustrated in Appendix A.4. The high-level planner is given a task description $\tau$, obtains environment state $M^t$ from EPM (Sec. 3.1), and selects an action (Navigate/Open/Pick/Place) parameterized by entities from $M^t$, which are translated into low-level controls via a skill policy. We use privileged skills to focus our study on perception and planning and note that our framework is agnostic to skill implementation. Skills execute until failure or termination, at which time the planner is called again. For a fair comparison, we adopt the same skills for the baselines.

**High-Level Planner**. Following Chang et al. (2025), we use ReAct (Yao et al., 2023) to enable an LLM to take actions in the environment. The prompt at time $t$ is concatenated text representing objects in $M^t$, where each object consists of a UID and a description from EPM (Sec. 3.2). Auto-regressive action predictions are interleaved with world representations, resembling an instruct LLM format (Grattafiori et al., 2024). To avoid excessive context lengths, at each step, only objects updated by EPM are added to the context. See Appendix A.4 for details and prompt examples.

### 4.2 Training the High-Level Planner

While the prompting format described above enables non-trival zero-shot performance from pretrained LLMs (see Table 3), the domain of embodied planning is underrepresented in typical LLM training data. The primary challenge is sourcing high-quality data for embodied planning to further improve performance. We procure this data in two ways: first, by transforming human demonstrations teleoperated in simulation into robot-compatible plans, and second, using a novel online reinforcement learning method to improve the planner with experience in the environment.

**Training With Human Demonstrations**. We use data derived from human demonstrations to train planning models to solve PARTNR tasks (Chang et al., 2025). The factorization of perception and action (Liang et al., 2023) poses the following challenge: how can we learn to plan from demonstrations when different systems of perception induce different optimal actions? For example,

---

Note that these heuristics are only used to generate training data, not at inference time.

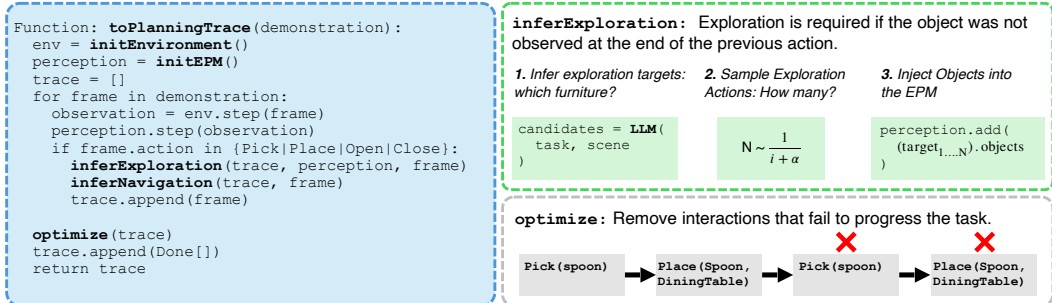

Figure 3: **Planning Traces from Human Demonstration**. We replay human demonstrations in simulation while rolling out a perception system in-the-loop. To adapt the trace to the perception system and teach the agent exploration, we infer a sequence of exploration actions. Finally, we mitigate sub-optimal human demonstrations by removing interactions that fail to make task progress.

an agent with "pixel-perfect" perception can simply spin in place to identify many objects in a room, whereas a nearsighted agent must move about the space to achieve similar understanding (Gibson, 1979). We address this problem by replaying teleoperations in simulation while running perception in-the-loop. This enables perception-specific planning traces without needing to collect perception-specific demonstrations. We fine-tune LLMs on these traces using LoRA (Hu et al., 2022).

Our approach to deriving training data for high-level planning is illustrated in Fig. 3. Each human demonstration consists of a sequence of agent and object poses along with an interaction label (`Pick`/`Place`/`Open`/`Close`/`None`). At each step, we update the environment state and EPM. We derive high-level planning actions as follows. Each interaction label implies a planning action. The trace must also reflect object discovery and navigation, so we add `Explore` and `Navigate` actions accordingly. In particular, if the EPM has not yet detected the interaction object by the end of the previous action, we add `Explore` actions to find it. To teach exhaustive search behavior, we sample exploration actions that fail to reveal the object. To teach robustness to EPM hallucinations, we sample Nav-Pick actions to hallucinated objects during exploration. Finally, we optimize the resulting trace by removing interaction sequences that fail to make progress toward success according to the episode's evaluation function. This enables us to mitigate occasional suboptimal behaviors in the human demonstration dataset. Collectively, this process yields a planning trace we use for training. We provide further details in Appendix A.6.

**Improving Planning With *DDAFT*.** To improve planning performance with online experience, we finetune our planner using a novel formulation of value-function-free RL for LLMs we call *Dynamic Difficulty-Aware FineTuning*, or *DDAFT*. The goal of *DDAFT* is to improve performance by efficiently searching the planning space for higher-quality planning behavior. While typically, methods focused on improving reasoning (Shao et al., 2024; Ethayarajh et al., 2024) sample instructions uniformly from the training data, *DDAFT* shapes the sampling space toward difficult instruction to improve sample efficiency. This is similar to DART-Math (Tong et al., 2024), however, DART-Math produces a static finetuning dataset, while *DDAFT* is run iteratively, using the current policy to dynamically estimate the difficulty of instructions, leading to superior performance and sample efficiency.

Given a set of tasks available for training $\tau \in \mathcal{T}$ and a task-conditioned initial planning policy $\pi_0(a|s, \tau)$ (e.g., the one we learned above from human demonstrations), we first roll out $\pi_0$ on all episodes in $\mathcal{E}$ producing an initial dataset of traces (denoted with $x$) with task rewards (denoted with $r$), $\mathcal{D}_0 = \{(x_0, r_0), \ldots (x_n, r_n)\}$. The process continues by alternating between finetuning a planning model on the dataset $\mathcal{D}_t$ and sampling new traces to form a new dataset. Many different loss functions commonly used in reinforcement learning or language model finetuning are compatible with this formulation (Schulman et al., 2017; Shao et al., 2024; Ethayarajh et al., 2024). For the experiments in this work we fit the RFT objective (Yuan et al., 2023).

Instead of sampling new traces uniformly across the training episodes, as is typically done for LLM reasoning finetuning (Shao et al., 2024; Yuan et al., 2023), *DDAFT* biases sampling so that exploration is focused on episodes for which the dataset currently contains no successful examples. We do this by sampling which episodes to generate new traces on from a distribution induced by the

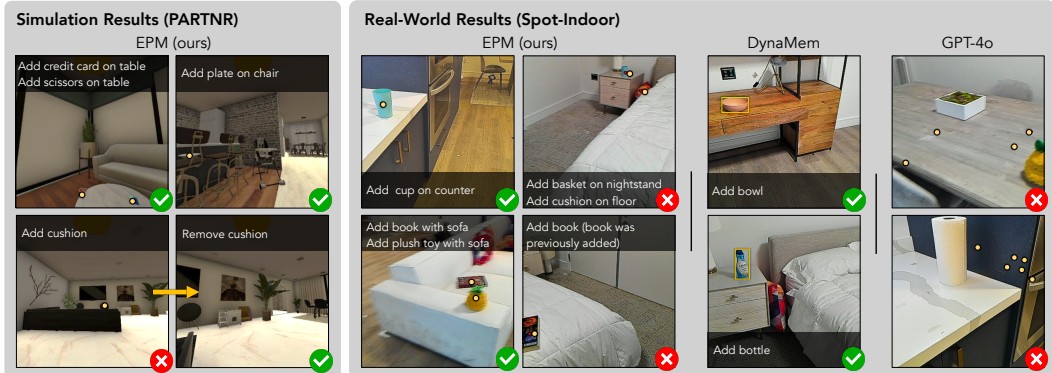

Figure 4: **Qualitative Perception Results. Left**: Offline evaluation results of the EPM on the PARTNR dataset, with examples of object addition (top) and removal of mislabeled objects (bottom). **Right**: Results for EPM, along with the proposed baselines in the real world. The orange dots and bounding box highlight the objects localized by the different methods.

softmax over the failure rates of each episode in the dataset so far. The formalism of this approach, along with comparisons against DART-Math and Online-RFT can be found in Appendix A.7.

## 5 EXPERIMENTS

We measure the ability of our system (1) to perceive and represent dynamic environments and (2) to enable agents to perform long-horizon planning. We first evaluate the EPM in egocentric videos, with annotated actions and camera poses, that depict a robot interacting with objects in an indoor environment. This experiment demonstrates perception capabilities of EPM independent of planning. Next, we present our main result by evaluating EPM within our planning framework on the PARTNR benchmark (Chang et al., 2025). This evaluation demonstrates the joint ability of EPM and our planning methods to perform long-horizon natural language tasks in large indoor environments.

### 5.1 PERCEPTION EVALUATION

We first evaluate the EPM independently of planning: given an agent taking actions in an environment, we measure the ability to track the environment over time. We consider two datasets:

- **PARTNR** (Chang et al., 2025): To evaluate the performance of our memory representation in large indoor environments, we generate single-agent demonstrations from the PARTNR validation set and obtain the environment state every $K$ agent steps, as described in Sec 3.2. This results in 1000 demonstrations in 12 different scenes with different object configurations in each episode.

- **Spot-Indoor**: To measure whether our representation transfers to real-world scenarios, we collect a dataset of 30 interaction sequences (15 human, 15 robot) recorded in a $\sim 155m^2$ apartment using a Spot robot. While large-scale datasets like EPIC Kitchens (Damen et al., 2020) and EPIC Fields (Tschernezki et al., 2023) provide dynamic scenes and global poses, they lack the dense depth data required for our approach. Our dataset contains 1,100 frames where, in each sequence, target and distractor objects are relocated between various receptacles. In half of the sequences, humans make changes to the environment as well. We label the dataset by manually annotating the relevant objects and distractors, listing the first frame where a given object is seen and the frames at which an object moves.

**Metrics.** To evaluate the spatial memory, we construct at each timestep $t$ an environment graph $\hat{G}_t$ given the environment state $M^t$. Nodes correspond to entities in $M^t$ (objects or furniture) and edges denote relations. We define a partially observable graph $G_t$ using groundtruth, representing the environment that can be observed given the agent's observations up to time $t$. We align $G_t$ and $\hat{G}_t$ with bipartite matching based on node distances (matching entities if the name matches and distance is $<0.5$) and report Precision, Recall and F1 scores for nodes and edges averaged across time.

**Baselines.** We compare our approach with the following external memory baselines:

| Method | Dataset | Node Metrics | | | Edge Metrics | | |
|---|---|---|---|---|---|---|---|
| | | Precision | Recall | F1 | Precision | Recall | F1 |
| GT | PARTNR | $0.86_{\pm0.01}$ | $0.54_{\pm0.02}$ | $0.60_{\pm0.02}$ | $0.83_{\pm0.01}$ | $0.54_{\pm0.02}$ | $0.63_{\pm0.02}$ |
| GPT-4o | PARTNR | $0.00_{\pm0.00}$ | $0.05_{\pm0.01}$ | $0.00_{\pm0.00}$ | $0.00_{\pm0.00}$ | $0.00_{\pm0.00}$ | $0.00_{\pm0.00}$ |
| DynaMem | PARTNR | $0.04_{\pm0.00}$ | $0.10_{\pm0.01}$ | $0.05_{\pm0.00}$ | - | - | - |
| EPM | PARTNR | $\mathbf{0.36_{\pm0.01}}$ | $\mathbf{0.42_{\pm0.01}}$ | $\mathbf{0.34_{\pm0.01}}$ | $\mathbf{0.66_{\pm0.02}}$ | $\mathbf{0.38_{\pm0.02}}$ | $\mathbf{0.46_{\pm0.02}}$ |
| GPT-4o | Spot-Indoor | $0.01_{\pm0.00}$ | $0.65_{\pm0.04}$ | $0.02_{\pm0.00}$ | $0.00_{\pm0.00}$ | $0.00_{\pm0.00}$ | $0.00_{\pm0.00}$ |
| Dynamem | Spot-Indoor | $\mathbf{0.42_{\pm0.03}}$ | $\mathbf{0.73_{\pm0.03}}$ | $\mathbf{0.51_{\pm0.03}}$ | - | - | - |
| EPM | Spot-Indoor | $0.30_{\pm0.05}$ | $0.33_{\pm0.05}$ | $0.24_{\pm0.03}$ | $\mathbf{0.28_{\pm0.06}}$ | $\mathbf{0.12_{\pm0.03}}$ | $\mathbf{0.16_{\pm0.04}}$ |

Table 2: **Perception-only Results.** Perception evaluation outside the planning loop.

- **GT**: To demonstrate the challenges of the PARTNR dataset, we design a privileged version of EPM, where objects are added based on a groundtruth sensor if they are within a distance range from the agent and occupy a minimal area of the agent's observation.
- **GPT-4o**: We design the memory as in EPM, but obtain update operations via GPT-4o.
- **DynaMem**: We choose DynaMem (Liu et al., 2024a) as our main baseline as it is open-vocabulary, shown to work with LLM planners in the real-world, and handles dynamic scenes. DynaMem stores aggregated object embeddings in a global voxel grid, along with RGB-D and pose data. For a query, it retrieves the closest voxel via feature-matching (Zhai et al., 2023) and then uses an object-detector (Minderer et al., 2023) on the associated frame to locate the object. If found, the bounding box and depth provide the 3D position. We query each frame using all object categories in our dataset. As object relationships require exhaustive pairwise queries, we omit edge metrics.

### 5.1.1 Perception-only Results

In this section, we report perception results in an offline setting (i.e., outside the context of planning). EPM outperforms baselines significantly in simulation (Table 2 top). However, the performance is far from saturation (F1-node 0.34), due to the challenging nature of the dataset, where objects are often partially visible or seen from far away (see GT performance). Our approach can localize and add objects to the memory, along with their relationships, and remove incorrectly-added objects (Fig. 4 left). To evaluate real-world generalization, we report the results on the Spot-Indoor dataset (Table 2 bottom). Note that our memory operations are learned using only simulation data. Generally, the model is able to localize objects and relationships in the real world, but often mis-categorizes objects and fails to associate multiple instances of the same object (Fig. 4 right). More analysis can be found in Appendix A.3. DynaMem obtains the best performance, but requires querying an object detector with every object from a vocabulary, which is unfeasible with large vocabularies. As we show later in our planning results, these limitations lead to significant drops in planning performance and efficiency compared to our model. Also, note that DynaMem cannot infer relationships (hence, no edge metric is reported). GPT-4o tends to hallucinate objects and labels, resulting in much lower node precision.

### 5.2 Planning Evaluation

We evaluate the planning framework with EPM memory against a benchmark of household tasks. We use this online evaluation to draw conclusions about the utility of EPM as a dynamic memory for task planning and the efficacy of our planner to successfully perform tasks given this system of perception. We evaluate against a single-agent version of the PARTNR benchmark (Chang et al., 2025), and report results against the held-out validation split of 1,000 episodes covering 12 unique scenes from the Habitat Synthetic Scenes dataset (HSSD (Khanna et al., 2024)).

**Metrics**. Following PARTNR evaluation protocols, we report metrics across axes of task completion and efficiency. *Success Rate*: The percentage of episodes completed successfully, *Percent Complete*: the average progress through each task, even if not fully completed, *Simulation Steps*: the average number of steps taken in the simulator across all episodes, *Planning Cycles*: the average number of high-level actions, and *Extraneous Actions*: the ratio of unnecessary rearrangements over total rearrangements. The last two are not relevant to the DynaMem baseline as it uses open-loop planning.

**Baselines.** We compare against the following baselines:

| Row | Method | Perception | Success Rate ↑ | Percent Complete ↑ | Simulation Steps ↓ | Planning Cycles ↓ | Extraneous Actions ↓ |
|---|---|---|---|---|---|---|---|
| 1 | DynaMem (Liu et al., 2024a) | GT | 0.17 ± 0.01 | 0.33 ± 0.01 | 7611 ± 130 | - | - |
| 2 | PP (Chang et al., 2025) | GT | 0.51 ± 0.02 | 0.67 ± 0.01 | **1570** ± 40 | **17.6** ± 0.4 | 0.19 ± 0.01 |
| 3 | PP+DDAFT (ours) | GT | 0.66 ± 0.02 | 0.78 ± 0.01 | 2160 ± 60 | 22.3 ± 0.5 | 0.19 ± 0.01 |
| 4 | HD (ours) | GT | 0.63 ± 0.02 | 0.74 ± 0.01 | 2110 ± 80 | 21.6 ± 0.6 | 0.17 ± 0.01 |
| 5 | HD+DDAFT (ours) | GT | **0.68** ± 0.02 | **0.80** ± 0.01 | 2550 ± 90 | 22.7 ± 0.5 | **0.15** ± 0.01 |
| 6 | DynaMem (Liu et al., 2024a) | DynaMem | 0.03 ± 0.01 | 0.11 ± 0.01 | 5090 ± 120 | - | - |
| 7 | PP (Chang et al., 2025) | Learned | 0.46 ± 0.02 | 0.65 ± 0.01 | **1850** ± 50 | **19.7** ± 0.4 | 0.28 ± 0.01 |
| 8 | PP+DDAFT (ours) | Learned | **0.58** ± 0.02 | **0.74** ± 0.01 | 2200 ± 60 | 23.2 ± 0.5 | 0.27 ± 0.01 |
| 9 | HD (ours) | Learned | 0.55 ± 0.02 | 0.69 ± 0.01 | 3040 ± 110 | 30.2 ± 0.8 | **0.17** ± 0.01 |
| 10 | HD+DDAFT (ours) | Learned | **0.58** ± 0.02 | **0.74** ± 0.01 | 2250 ± 70 | 23.6 ± 0.6 | 0.23 ± 0.01 |

Table 3: **Planning Results.** Results on the PARTNR single-agent benchmark. We pair planners with various forms of memory in the groundtruth (top) and learned (bottom) perception settings.

- **DynaMem**: The perception system described in Sec. 5.1 is combined with a grid-based navigation system that linearly combines voxel-last-seen timestamps and voxel-to-query alignment scores to identify exploration frontiers. The system explores until input query is found then picks or places on it. Based on proposed system in Liu et al. (2024b), we add a few-shot open-loop LLM planner to decompose tasks into pick-and-place objectives. We tested several ablations of DynaMem specifically testing performance if we used: (a) GT object segmentation and categories instead of object-detector, (b) room-labels along-with object categories (referred to as Room in appendix), and (c) only voxel-last-seen timestamp (TimeOnly) for exploration instead of linear combination. We report the best performing variant in Table 3. See Appendix A.5 for details.
- **PP**, PARTNR-Pretrained (Chang et al., 2025): LLM planning with EPM using the prompting strategy described in Section 4.1 using `Llama3.3-70b-Instruct`, with no additional finetuning.
- **HD**: Planning with EPM with `LLama3.1-8b-Instruct` fine-tuned on traces derived from **H**uman **D**emonstrations as described in Section 4.2. For training, we use demonstrations from the 100,000 tasks in the PARTNR Train split (Chang et al., 2025).
- **PP+*DDAFT***: Starting with the PP model as the base model, we run 5 iterations of *DDAFT* on a 2,000 episode subset of the PARTNR training data, sampling 2,000 episodes per iteration.
- **HD+*DDAFT***: Planning with EPM with a model fine-tuned on demonstration and RL data. Using HD as the base model, we perform policy improvement following PP+*DDAFT*.

### 5.2.1 PLANNING RESULTS

We report PARTNR planning results in Table 3 with groundtruth (top) and learned (bottom) perception and highlight key findings below.

**HD plans effectively from EPM representations.** Our HD method outperforms PP in the GT perception setting by 0.12 success (row 4 vs 2) despite a smaller model size (8b vs 70b parameters). Following *DDAFT*, this trend continues (0.68 vs 0.66 SR; row 5 vs 3). This shows that deriving traces from demonstrations is an effective approach to learn embodied task planning.

**HD adapts to EPM failures.** With learned perception, HD outperforms PP (row 8 vs 7) and even outperforms PP with perfect perception (row 8 vs 2). This shows that demonstrations teach effective planning without the need for collecting perception-specific planning traces.

***DDAFT* enables policy improvement over both PP and HD policies.** With GT perception, *DDAFT* leads to success improvements of 0.15 for PP and 0.05 for HD (comparing row 3 to row 2, and row 5 to row 4, respectively). This extends to learned perception, where we observe improvement of 0.12 and 0.03 success for PP and HD, respectively (row 8 vs 7, row 10 vs 9). This showcases the generality of *DDAFT* to improve text-based planning policies that leverage the EPM.

**Our methods with EPM in the loop outperform strong baselines in task planning.** Our methods (rows 8-10) outperform strong baselines (rows 6&7) by a large margin (55% and 12% improvement in success rate over DynaMem and PP, respectively). To ensure that DynaMem's lower performance is not attributable to simulation visuals, we also provide DynaMem with groundtruth perception (row 1). Nevertheless, its performance remains inferior to ours. In addition to exploration challenges, tasks may require specific instance or multiple instances of objects ('all forks') and furniture ('table in office') which are not handled well by the retrieval-then-detection based algorithms. It should be noted that our method is 3.5x faster than DynaMem. PP shows lower sim steps and planning cycles

than our methods. That is primarily due to solving shorter horizon tasks. See Appendix A.8-A.10 for more results and a further discussion.

### 5.2.2 REAL WORLD EVALUATION

To evaluate the effectiveness of our method in real-world scenarios, we test the PP+DDAFT planner together with the learned EPM in an indoor environment, using the Spot robot. We use our planning and memory system to predict high-level actions given a language instruction and adapt the work from Yokoyama et al. (2023) to parse the high-level instructions into motor skills. We test the system in 20 different scenarios, which require the robot to move objects to target locations in presence of clutter. The robot succeeds in 55% of the tasks, with failures coming from both planning and skill execution errors. In 70% of the tasks, the EPM succeeds at predicting the correct plan. More details can be found in the Appendix A.10.

## 6 CONCLUSION AND LIMITATIONS

We present EPM, a novel learnable memory for embodied planning models. We show that the explicit world representation of EPM allows for seamless integration with LLM-based planners. Further, we show that signal for training planning models on this representation can be extracted from general human demonstrations, and our novel *DDAFT* method improves planning performance in all settings. Our proposed planning approach outperforms baselines even when they are provided with ground-truth perception.

Our training is limited to simulation data due to the complex nature of collecting a large-scale real-world dataset for dynamic scenes with detailed planning actions. We leave real-world training for future work. We also note that purely text-based representations could inhibit planning performance in certain settings. Our EPM formulation is general enough to incorporate visual information as continuous features, provided the training data includes such modalities. For example, if a task requires reasoning about object properties not represented in text, the planner has no way to solve the task. In those scenarios, a hybrid system could pair the EPM with specialized visuomotor or VLA modules that operate directly on the current observation and do not require persistent memory. We leave this exploration for future work.

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
