# A APPENDIX

This paper introduces EPM memory for embodied planning, integrates this memory with LLM-based planners, and shows that such planners can be trained via novel methods involving human demonstrations and policy improvement (*DDAFT*). This appendix provides additional details on these contributions and is organized as follows:

We also include our source code, with instructions to generate data to train our planner as well as the EPM, as well as instructions to run the EPM in isolation and together with the planner.

## A.1 RELATED WORK: SEMANTIC SLAM

Several methods build dynamic scene representations within the context of simultaneous localization and mapping (SLAM) (Jiang et al., 2024; Wu et al., 2024; Fan et al., 2022; Yu et al., 2018). These fall in two major classes: NeRF-based SLAM (Jiang et al., 2024; Wu et al., 2024) and Semantic SLAM (Fan et al., 2022; Yu et al., 2018). Generally, these methods focus on optimizing the agent and camera poses in the presence of dynamic objects, rather than tracking these dynamic objects. Thus, it is not trivial to adapt these methods for embodied planning, since it would require tracking the seen objects, rather than removing them from the representation. DS-SLAM [4] tracks dynamic objects by building a semantic map of the environment, organized in an octo-tree map. To improve robustness, it optimizes the reconstructed scene by combining the output of an optical flow and semantic segmentation network with geometric constraints. However, it is limited to 20 semantic classes, and leveraging the octo-tree representation for planning remains an open question.

## A.2 EMBODIED PERCEPTION MEMORY DETAILS

We provide more details about our proposed EPM, including a description of the model, how the update operations are parsed and processed and how we train the model with simulation data.

### A.2.1 MODEL DETAILS

The EPM is composed of an entity list, which stores each of the entities in the environment and a VLM, which is trained to output operations to modify the entity list. We describe each of the components below.

**Entity List**. Entities in the memory store information about furniture and objects in the environment. Each entity contains a numerical identifier, a 3D coordinate locating the entity in the world frame, the distance from it was seen with respect to the agent, and a description, containing information about the entity and its state. Additionally, we store for each entity a dictionary of metadata, with information on whether this entity is currently picked by an agent or not. The Entity List is initialized with the furniture in the environment, with the 3D coordinate corresponding to the furniture centroid and the description corresponding to the furniture name. Entities are added, removed or updated based on the operations described in Sec. 3.2, which are predicted by the VLM.

**VLM**. We use a VLM to predict `Add`, `Remove` or `Update` operations. We use `LLaVa-OneVision-7b-Qwen2` as our base VLM, which we finetune with a low-rank adapter (Hu et al., 2022). The model takes as input a $(384 \times 384)$ RGB image, corresponding

to a robot's egocentric observation close to the robot gripper, and a text description containing the following information:

- **Entity List Content:** A text representation of the list of entities. For each entity, the text contains information about its 3D coordinate, description, and distance from where it was last seen. If the entity is currently being picked by the agent, the 3D coordinate is replaced by the text *"picked"*.
- **Last Action:** The last high-level action taken by the agent.
- **Egocentric Coordinates:** A list of 64 coordinates, obtained by uniformly sampling an $8 \times 8$ grid from the egocentric observation and converting them to the world frame via the depth information and camera pose.

We show on the next page an example prompt for the VLM.

### A.2.2 UPDATE OPERATIONS

We provide further details about the memory operations, and how they update the entity list.

- `Add (<coords>):<description>`: adds a new entity into the entity list. The description is parsed from the `<description>` field, and the 3D coordinate is derived by unprojecting `<coords>`. We also set the distance from which the new entity was seen by computing the depth at those coordinates. The numerical identifer is equal to the number of entities that have been added plus 100.
- `Remove k`: As described in the main paper, this operation removes an entity with identifier k.
- `Update k (<coords>):<description>`: Updates the description of the entity k. If the description contains the text "picked", it strips that word out and sets that object's pick state to `True`. If the description contains "placed", it strips the word out and sets the pick state to `False`. Additionally, it updates the coordinate of the object by unprojecting `<coords>`. And sets the depth as in the `Add` operation.

**Embodied Perception Memory Input**

```
Given your previous memory for the environment and your current observation, predict how
to update the memory for the environment.Your current observation has the following
coordinates:
(-10.6, 1.7, -3.8), (-10.8, 1.8, -3.8), (-11.0, 1.9, -3.8), (-10.8, 0.9, -4.7), (-10.8,
0.9, -4.7), (-12.3, 2.8, -3.9), (-12.7, 2.8, -4.2), (-12.7, 2.5, -4.5), (-10.6, 1.5,
-3.8), (-10.8, 1.6, -3.8), (-11.0, 1.7, -3.8), (-11.3, 1.8, -3.8), (-11.7, 2.1, -3.8),
(-12.5, 2.4, -3.8), (-12.7, 2.3, -4.2), (-13.9, 2.8, -4.3), (-10.6, 1.3, -3.8), (-10.8,
1.3, -3.8), (-11.0, 1.4, -3.8), (-11.3, 1.5, -3.8), (-11.7, 1.6, -3.8), (-12.4, 1.9,
-3.8), (-12.7, 1.8, -4.1), (-15.6, 2.8, -4.0), (-10.7, 1.1, -3.8), (-10.8, 1.1, -3.8),
(-11.0, 1.2, -3.8), (-11.3, 1.2, -3.8), (-11.7, 1.3, -3.8), (-12.4, 1.4, -3.8), (-12.7,
1.4, -4.1), (-16.2, 2.0, -3.8), (-10.7, 0.9, -3.8), (-10.8, 0.9, -4.0), (-11.0, 0.9,
-3.8), (-11.3, 0.9, -3.8), (-11.7, 0.9, -3.8), (-12.3, 0.9, -3.8), (-12.7, 0.9, -4.0),
(-15.9, 0.9, -3.8), (-10.7, 0.8, -4.1), (-10.8, 0.8, -4.2), (-10.9, 0.8, -4.3), (-11.0,
0.8, -4.3), (-11.2, 0.7, -4.3), (-12.3, 0.5, -3.8), (-12.7, 0.5, -4.0), (-15.3, 0.1,
-3.8), (-10.7, 0.6, -3.8), (-10.9, 0.5, -3.8), (-11.0, 0.5, -3.9), (-11.3, 0.4, -3.8),
(-11.2, 0.6, -4.3), (-12.2, 0.1, -3.8), (-12.7, 0.0, -4.0), (-13.2, 0.0, -4.2), (-10.7,
0.4, -3.8), (-10.9, 0.3, -3.8), (-11.1, 0.3, -3.8), (-11.2, 0.3, -4.0), (-11.2, 0.5,
-4.3), (-11.9, 0.0, -4.0), (-12.1, 0.0, -4.2), (-12.4, 0.0, -4.4)
Your previous action is: Pick[Object.149]
Your previous memory is:
Object 100 (-2.9, 0.2, -2.1): table
Object 101 (-3.1, 0.5, -8.8): washer dryer
Object 102 (-4.7, 0.5, -8.8): washer dryer
Object 103 (-18.5, 0.4, -9.0): table
Object 104 (-5.0, 1.1, -7.5): shelves
Object 105 (-0.6, 0.4, 3.0): chair
Object 106 (-13.5, 0.4, -7.0): chair
Object 107 (-2.8, 0.5, -0.6): couch
Object 108 (-12.4, 0.4, -2.3): counter
Object 109 (-8.6, 0.5, -2.8): chair
Object 110 (-15.9, 0.9, -9.4): bed
Object 111 (-4.1, 0.3, 5.7): table
Object 112 (-12.2, 0.4, -8.6): chair
Object 113 (-14.5, 0.3, -10.3): table
Object 114 (-17.3, 0.3, -10.3): table
Object 115 (-0.3, 1.3, -3.2): shelves
Object 116 (-19.7, 0.4, -6.6): chair
Object 117 (-4.9, 0.5, -1.3): chair
Object 118 (-0.7, 0.5, -1.5): chair
Object 119 (-10.5, 0.4, -4.1): table
Object 120 (-13.6, 0.2, -9.0): stool
Object 121 (-6.8, 0.4, -7.7): table
Object 122 (-6.0, 0.4, -7.7): chair
Object 123 (-6.5, 0.4, -7.1): chair
Object 124 (-7.1, 0.4, -7.1): chair
Object 125 (-7.7, 0.4, -7.7): chair
Object 126 (-10.5, 0.4, -8.7): table
Object 127 (-12.3, 0.4, -0.5): cabinet
Object 128 (-12.3, 0.4, -3.2): cabinet
Object 129 (-9.1, 0.5, -1.8): chair
Object 130 (-9.1, 0.5, -1.3): chair
Object 131 (-9.1, 0.5, -2.3): chair
Object 132 (-15.9, 0.2, -8.0): bench
Object 133 (-2.7, 0.8, 5.0): bed
Object 134 (-1.3, 0.3, -0.8): table
Object 135 (-4.4, 0.3, -0.8): table
Object 136 (-11.6, 0.2, -8.1): bench
Object 137 (-1.6, 0.2, 3.0): bench
Object 138 (-8.6, 0.4, 1.8): table
Object 139 (-0.3, 0.2, -7.8): bench
Object 140 (-6.7, -0.0, 5.9): chest of drawers
Object 141 (-9.6, 0.0, -1.8): counter
Object 142 (-2.5, 0.0, 0.3): chest of drawers
Object 143 (-2.8, 0.0, -3.5): stand
Object 144 (-5.6, -0.0, 0.8): chest of drawers
Object 145 (-12.4, 0.0, -1.3): cabinet
Object 146 (-11.3, 0.0, -3.3): fridge
Object 147 (-13.1, 0.0, -9.0): table
Object 148 (-16.7, 0.0, -4.0): chest of drawers
Object 149 (-10.7, 0.9, -4.2) seen from 0.4 m: candle with the Object.119
Object 150 (-10.8, 0.9, -4.0) seen from 0.7 m: candle holder with the Object.119
Your next update is:
```

Table 4: Hyperparameters used for training the EPM.

| Hyperparameter | Values |
|---|---|
| Learning Rate | $2e^{-5}$ |
| LR Schedule | Linear Decay with warmup |
| LoRA $\alpha$ | 128 |
| LoRA Rank | 64 |
| LoRA Dropout | 0.01 |
| LoRA Params | all linear projections for both the vision backbone and language model |
| Batch Size | 64 |
| Training Duration | 4 Epochs |

### A.2.3 TRAINING

To train the EPM, we generate a dataset of memory updates on the PARTNR benchmark (Chang et al., 2025). For this, we use a heuristic planner to obtain single-agent trajectories on the PARTNR training set. After filtering unsuccessful tasks, and tasks with object state changes, we end up with 71,864 training episodes. For every episode we sample one of every 10 steps and use privileged information and heuristic to derive updates for the memory. In particular, for a given step, we use the following rules:

- We add an `Add` operation for every object in the current observation that is not currently in the list of entities, occupies at least 0.001% of the screen and is closer to 6 meters from the agent. In such case, the 2D coordinate is computed by drawing a bounding box around the object mask and computing the center. The description is set as the category of the object and additionally the text "with the Entity.X" if the furniture where the object is placed is visible.

- If an object already in the EPM is currently being picked, placed, open or closed or we can see the furniture where it is located, we add an `Update` operation. We set the coordinates to be the location of the object and the description to contain the text "picked", "placed", "open", "closed" if an action has been applied to it.

- To train the EPM to remove objects, we modify 1% of the steps to include remove operations. For this, we sample a 3D coordinate visible from the agent's current egocentric view, and add a random object into the EPM at that coordinate, with a depth larger than the current's coordinate depth. Then we add a `Remove` operation to delete that synthetically added object.

- If none of the previous conditions hold, we label that state as `No updates`.

Given that this process results in a big majority of steps being `No updates`, we balance the dataset to contain the same proportion of `No updates`, `Update` and `Add`, keeping removals at 1%, resulting in a dataset of 569,033 samples. We train the EPM via supervised finetuning on this dataset, using a low-rank adapter. The model is trained for 4 epochs on 8 NVIDIA H100 GPUs, during 74 hours. The training hyperparameters are listed in Table 4.

### A.3 SPOT-INDOOR PERCEPTION ANALYSIS

From Table 2, we dive deeper in three areas of the perception results of EPM on Spot-Indoor. First, we ignore object class names when evaluating EPM relative to ground truth. This is motivated to measure the ability to detect "objectness" over time. We observe precision and recall increasing from 0.30 and 0.33 to 0.37 and 0.42, respectively. This experiment shows that there is room to improve object detection.

Second, we set the matching threshold between EPM entities and ground truth entities to infinity to consider object class names exclusively. This is motivated to understand the ability to detect and classify independent of localization. We observe precision and recall increasing from 0.30 and 0.33 to 0.39 and 0.42, respectively. This experiment shows that the detection limitations are not solely localization errors.

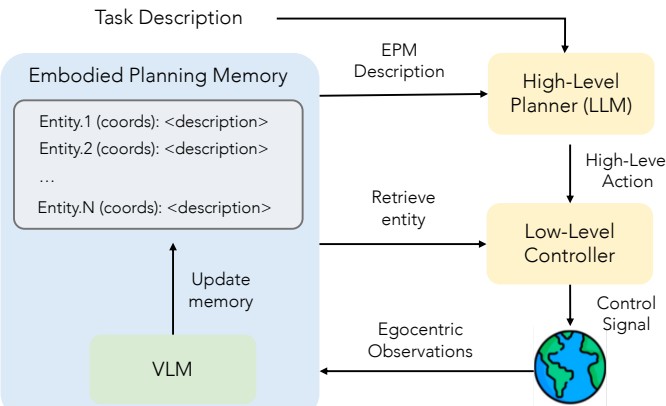

Figure 5: **Planner Overview.** The High-Level Planner obtains an EPM description, representing the last updates and the description of the task, and produces a high-level action, consisting of a verb and a list of entities. The Low-Level controller uses the EPM information to ground those entities in the environment, and produce low-level actions. As the agent interacts in the environment, the observations are used to update the EPM.

Finally, we count the number of predicted entities that are unmatched to ground truth entities (false positive detections). This is motivated to understand the propensity of EPM to hallucinate objects. We ignore object class names during matching to control for misclassifications. On average, we find 3.6 hallucinations per frame and 6.4 hallucinations in the final state. This experiment shows that hallucination reduction is one axis of future improvement.

## A.4   PLANNING DETAILS

We provide more details of the planner, including how it is connected with the EPM and how it is prompted.

Figure 5 shows an overview of our system. The EPM provides a text description of the environment, consisting of the entities in the cache at the first step and the EPM updates after that. Given this description, the task description, and the last action taken by the agent, the planner produces a high-level action, consisting of a verb $u \in$ {Navigate, Pick, Place, Open, Close, ExploreFurniture} and a sequence of EPM entities. Every 10 steps, the agent's egocentric observations are used by the EPM to update the environment representation.

In each planning step, one of the skills above is invoked with its necessary arguments referencing objects in the EPM (e.g. Navigate[Object.193], Pick[Object.210]). For each object which is an argument to a skill, the system finds the object in the simulation that is nearest to the stored 3d coordinate of said object. This is determined by $L^2$ distance from the 3d coordinate in the memory and the nearest point on the surface of simulation objects. After the nearest simulation object has been determined for each argument, we invoke the oracle version of the skill as defined in PARTNR (Chang et al., 2025) using the simulation objects. If any argument does not have a simulation object within one meter the skill is not executed and an error message is returned to the planner.

To reduce context growth, we only add changes to the EPM into the planner context. If the entire entity list from the object memory is added instead, success rates drop significantly (i.e SR falls from 0.46 to 0.32 for the PARTNR planner with EPM memory). Using only the changes to EPM in the planner context causes the planner context to grow by only 32.8 tokens per skill call on average, meaning even episodes which run until timing out (50 skill calls) produce planner contexts less than 3500 tokens. This is easily within the training context length for modern models.

An example of the prompting strategy in the form of a full trace is reproduced below. The system header is kept the same across all tasks. Changes to the text descriptions of EPM change are interleaved with actions predicted by the model autoregressively.

---

**Embodied Perception Memory Planning**

```
<|start_header_id|>system<|end_header_id|>

You are an agent that solves multi-agent planning problems. The task assigned to you
will be situated in a house and will generally involve navigating to objects, picking
and placing them on different receptacles to achieve rearrangement. You strictly follow
any format specifications and pay attention to the previous actions taken in order to
avoid repeating mistakes.

You should be close to an object or a furniture to interact with it. That means you
should navigate next to the object or furniture to make sure you are close enough.

You are playing the role of the task receiver. This means if the instruction says
something like "You should move the object and I will wash it", then you should move the
object and the other agent should wash it.

Many calls to the same action in a row are a sign that something has gone wrong and you
should try a different action.<|eot_id|><|start_header_id|>user<|end_header_id|>

Task: Move the plant container from the living room table to the bedroom shelves.

This is the list of objects seen so far, together with their coordinates:
Object.100: washer dryer in bathroom_2
Object.101: table in living_room_1
Object.102: shelves in bedroom_1
Object.103: table in bedroom_2
Object.104: shelves in bedroom_1
Object.105: stool in bedroom_1
Object.106: table in kitchen_1
Object.107: bed in bedroom_1
Object.108: couch in living_room_1
Object.109: bed in bedroom_2
Object.110: table in living_room_1
Object.111: cabinet in kitchen_1
Object.112: fridge in living_room_1
Object.113: stand in bedroom_2

Use the description of the scene to plan actions efficiently. Reason about what you have
seen so far and how close things are.

Possible Actions:
- Close: Used for closing an articulated entity. You must provide the name of the
furniture you want to close. Example (Close[Object.119])
- ExploreFurniture: Search a specific furniture by navigating to it, facing it, and
looking around a bit. The input to the skill is the EXACT name of the furniture to be
visited. Use the furniture names provided in the house description. This tool does not
open or close articulated links (e.g. drawers, cabinets). API template:
ExploreFurniture[Object.251]
- Navigate: Used for navigating to an entity. You must provide the name of the entity
you want to navigate to. Example (Navigate[Object.412])
- Open: Used for opening an articulated entity. You must provide the name of the
furniture you want to open. Example (Open[Object.119])
- Pick: Used for picking up an object. You must provide the name of the object to be
picked. The agent cannot hold more than one object at a time. Example (Pick[Object.132])
- Place: Used for placing an object on a target location. You need to provide the name
of the object to be placed, the name of the furniture where it should be placed, spatial
relation ("on" or "within") describing the relation between the object and furniture.
The object to be placed must already be held by the agent (i.e. picked previously). In
addition to these, you can request to place the object near another object. For that you
can optionally provide a spatial constraints ("next_to") and the name of the reference
object. To place next to an object, the reference object must already be on the target
furniture. API template - Place[<object_to_be_placed>, <spatial_relation>, <furniture to
be placed on>, <spatial_constraint>, <reference_object>]. spatial_constraint and
reference_object should be set to "None" when necessary.
- Done: Used to indicate that the agent has finished the task. Example (Done[])
```

---

```
What is the next action to make progress towards completing the task?
Return your response in the following format

<next action call>
Assigned!

Here is an example:
Navigate[<object>]
Assigned!
<|eot_id|><|start_header_id|>assistant<|end_header_id|>

Navigate[Object.101]
Assigned!<|eot_id|><|start_header_id|>user<|end_header_id|>

Result: Successful execution!
The following objects were updated:
Add Object.114: cushion with the Object.110
Add Object.115: plant container with the Object.101
Add Object.116: vase with the Object.101
Add Object.117: box with the Object.108
<|eot_id|><|start_header_id|>assistant<|end_header_id|>

Pick[Object.115]
Assigned!<|eot_id|><|start_header_id|>user<|end_header_id|>

Result: Successful execution!
The following objects were updated:
Update Object.115 (picked): plant container
<|eot_id|><|start_header_id|>assistant<|end_header_id|>

Navigate[Object.102]
Assigned!<|eot_id|><|start_header_id|>user<|end_header_id|>

Result: Successful execution!
No object updates.
<|eot_id|><|start_header_id|>assistant<|end_header_id|>

Place[Object.115, on, Object.102, None, None]
Assigned!<|eot_id|><|start_header_id|>user<|end_header_id|>

Result: Successful execution!
The following objects were updated:
Update Object.115: plant container with the Object.102
<|eot_id|><|start_header_id|>assistant<|end_header_id|>

Done[]
Assigned!<|eot_id|>
```

## A.5 PLANNING WITH DYNAMEM: DISCUSSION AND ABLATIONS

**Planning.** DynaMem (Liu et al., 2024a) uses the system setup from Liu et al. (2024b) to perform planning experiments with open loop planning for a single object. We take a similar approach where we plan for the entire task by breaking down given PARTNR task into multiple pick-and-place primitives; similarly to their approach, this means errors and faults are not handled by the planner. However, to provide the stack with a fair chance of attempting rearrangement for every object, we do not report task failure when an object is reported undiscovered. Instead, we progress the plan to the next pick action. If the stack reports failure trying to find the right receptacle, we simplify the problem by giving it a 'place on any table' action, such that we can roll out the entire episode. This allows our task completion percentage metric to better capture per-object rearrangement success for DynaMem stack (scaled by length of episode). For prompting this planner, we use the prompt provided in DynaMem GitHub code (Robot, 2025) and edit it to plan with just the pick and place primitives. Each primitive is defined such that it first searches the entire house for provided input and then picks it up (or places held object on it) once found. We use the simulation setup presented in PARTNR (Chang et al., 2025) to implement these experiments. We use object and context provided by the planner to look up relevant frames using SigLIP and use just the object-name for detection via OWL-ViT2 or GT object detector.

**DynaMem Ablations.** As described in Section 5.2 we test DynaMem ablations across three independent variables against the PARTNR benchmark. For DynaMem-GT, we use ground-truth object segmentation masks along with categories, as obtained from simulator, as input to the voxelization process. The textual label of the object is encoded as SigLIP feature and ascribed to the masked

| Row | Method | Perception | Success Rate ↑ | Percent Complete ↑ | Simulation Steps ↓ |
|---|---|---|---|---|---|
| 1 | DynaMem-Room-TimeOnly | GT | $0.17 \pm 0.01$ | $0.33 \pm 0.01$ | $7611 \pm 7$ |
| 2 | DynaMem-Room | GT | $0.10 \pm 0.01$ | $0.20 \pm 0.01$ | $5528 \pm 110$ |
| 3 | DynaMem | GT | $0.07 \pm 0.01$ | $0.15 \pm 0.01$ | $5520 \pm 130$ |
| 4 | HD+DDAFT (ours) | GT | $0.68 \pm 0.02$ | $0.80 \pm 0.01$ | $2550 \pm 90$ |
| 5 | DynaMem | DynaMem | $0.03 \pm 0.01$ | $0.11 \pm 0.01$ | $5090 \pm 120$ |
| 6 | HD+DDAFT (ours) | Learned | $0.58 \pm 0.02$ | $0.74 \pm 0.01$ | $2250 \pm 70$ |

Table 5: **Planning Results.** Online planning results on the PARTNR single-agent benchmark for all DynaMem ablations against our best-performing planning algorithm with both GT and learned perception (mean and standard error)

point-cloud of that object. For adding room-labels DynaMem-GT, we pass the room that the object is in as another textual label to the SigLIP encoder, e.g. "lamp in bedroom". This way each voxel has an averaged feature corresponding to object's category and the room it is in. This information is provided from privileged source. This variant puts DynaMem's a priori information at par with the information available to the EPM, i.e. all furniture as well as the layout of the house. Finally, since PARTNR houses are much larger and complex than DynaMem environments, for testing DynaMem-GT-Room with last-seen timestamp as the only guide to exploration (TimeOnly) we modulate the weights of time-heuristics and alignment-heuristic to get final values for frontier priority. One of them, DynaMem-GT was described in Section 2. Planning with all these variations is done same as described in previous passage.

**Results.** We see steady improvement in DynaMem's performance on PARTNR as we increase the amount of privileged information given to it and when we only use last-seen timestamps for exploration, see Table 5. Adding privileged object segmentation information to DynaMem results in an improvement of 4pp in percentage episode success metric and 4pp in binary success of solving episodes. Adding room labels (akin to object instance information) results in another improvement of 5pp in percentage success metric. Finally, if we keep GT object instance information but use only last-seen timestamps for exloration we find the best performing variant of DynaMem with overall percentage success rate of 0.33 and episode success rate of 0.17.

**Discussion.** Given the specificity of PARTNR tasks, we observe that DynaMem has a controllability problem. Using SigLIP features does not provide enough context for the perception component to disentangle between moving a picture frame that is on a table to a picture frame hanging on the wall next to a table. We note room-layout to be an important feature for solving PARTNR tasks, however we still see problems with exploration with "object or receptacle undiscovered" being the primary reason for episode failures. Our results suggest the EPM natively does a better job of encoding edge relations and preserving explicit context which supports overall better planning performance. DynaMem's applicability to large-scale, multi-room scenarios, like in PARTNR, also suffers when we use a linear combination of feature alignment and voxel's last-seen timestamp to prioritize exploration frontiers. On the other hand, if we take the alignment-value out of the equation and give the algorithm sufficient steps to explore the environment, we see a huge improvement in performance. This indicates the combination of time as well as feature alignment for prioritizing frontier exploration is useful in smaller environments but not in large-scale houses or tasks as seen in PARTNR.

---

**Example PARTNR Task Input and Output**

```
Instruction:
Move the electronic cable, sushi mat, and toy pineapple from the living room to the
bedroom. Place the electronic cable on the table and the sushi mat and toy pineapple on
the tables.

Planned actions:
  'Pick[electronic cable in living room]',
  'Place[bedroom table]',
  'Pick[sushi mat in living room]',
  'Place[bedroom tables]',
  'Pick[toy pineapple in living room]',
```

```
    'Place[bedroom tables]',
    'Done[]'
```

---

**Planning prompt for DynaMem**

```
You are a helpful planning agent who helps with household tasks. Your job is to break
down the given instruction into a list of actions. Each subtask should be a single
action that can be executed by the agent. The agent only understands following actions:
- Pick[object-name, context]
- Place[furniture-name, context]
- Done[]
If there is no additional context to the object then only use the object-name do not add
any
unspecified context. The object's name should always be the first in this list.
The agent can only pick and place objects. It cannot move them directly. The agent can
only pick up one object at a time. The agent can only place the object in one location
at a time.

Following are a few examples of how to break down the instruction into subtasks:

Example 1:
Instruction: Help me move the duck toy from the chest of drawers in play room to the
living room.
Actions:
- Pick[duck toy, chest of drawers in play room]
- Place[table, living room]
- Done[]

Example 2:
Instruction: Let's decorate! Move the vase and the plate from the kitchen to the bedroom.
Actions:
- Pick[vase, in kitchen]
- Place[bed, in bedroom]
- Pick[plate, in kitchen]
- Place[bed, in bedroom]
- Done[]

Example 3:
Instruction: I need a clean-up in this house. Move the apple to the kitchen counter and
the banana to the fridge.
Actions:
- Pick[apple]
- Place[kitchen counter]
- Pick[banana]
- Place[fridge]
- Done[]

Input:
Instruction: {instruction}
Actions:
```

## A.6    ADDITIONAL DETAILS AND EXPERIMENTS FOR HUMAN DEMONSTRATION TRAINING

### A.6.1    IMPLEMENTATION DETAILS

Referring back to Figure 3, we describe the following components involved in deriving planning traces from human demonstrations:

**Inferring Exploration.** Exploration is required if, by the end of the previous action, the picked object has not yet been observed by the perception system. Thus, we add an action to explore the furniture containing the object. However, a rational agent cannot always determine the exact furniture to explore from the instruction and past actions alone. For example, the instruction *"Bring the cup from the living room back to the kitchen counter."* implies that the cup will be found in the living room, but not if the cup is on a table or the couch. Thus, we teach the planner to explore a rational sequence of furniture until finding the object. First, an LLM determines the set of candidate furniture with which the object could be found. We then reduce this set of candidates to those that have not yet been explored in the trace. Then, we sample how many of these candidates should appear in the training trace. The probability of sampling $i$ candidates is given by $w_i = \frac{1}{i+\alpha}$ normalized by $\sum_{j=1}^{N} w_j$ where $i \in \{0...N\}$, $N$ is the number of candidates, and $\alpha$ controls the skew toward

Table 6: Hyperparameters used for training planners from demonstration data.

| Hyperparameter | Values |
|---|---|
| Learning Rate | $2.5e^{-4}$ |
| LR Schedule | Linear Decay with warmup |
| LoRA $\alpha$ | 128 |
| LoRA Rank | 32 |
| LoRA Dropout | 0.01 |
| LoRA Params | query and value projections |
| Batch Size | 10 |
| Training Duration | 30 Epochs |

sampling fewer candidates. We set $\alpha = 2.0$ empirically. For each chosen exploration target, we inject associated objects into the EPM that have not already been observed.

**Injecting Hallucination Actions.** The EPM occasionally hallucinates object instances that are relevant to the task. Given our environment representation, the only way a planner can determine if an object is real or hallucinated is by attempting to interact with it and receiving a skill response. Thus, we sample Nav-Pick action sequences to hallucinated instances of objects that the human demonstration actually interacted with. We inject these actions during the exploration phase prior to the human-successful Pick action. To maintain reasonable-length planning traces, we limit the sampling of such actions to a probability $p = 0.10$. The desired behavioral result is that the model will learn to keep searching for task-relevant objects when hallucinations are encountered.

**Inferring Navigation.** Navigation actions are inserted to precede all interaction events (`Pick` / `Place` / `Open` / `Close`), with the target set to the centroid of the object or the furniture to be interacted with.

**Optimizing the Trace.** Human demonstrations may contain spurious or non-optimal object interactions. Once the trace has been derived in full, we remove unnecessary actions by de-duplicating action sequences (Fig. 3, right). We also remove sequences that fail to progress the step-wise delta of the episode's evaluation function Chang et al. (2025).

We train planning models on the resulting traces. Specifically, we fine-tune `Llama-3.1-8b-Instruct` Grattafiori et al. (2024) with LoRA Hu et al. (2022). Each model is trained on 8 NVIDIA A100 GPUs, during 72 hours. Deriving training traces from human demonstrations with EPM in-the-loop used 72 NVIDIA A100 GPUs during 48 hours. Training hyperparameters are in Table 6 and ablations of the method are presented in Section A.6.2.

### A.6.2 ABLATION EXPERIMENTS

In Table 7, we validate design choices that support the derivation of perception-specific planning traces from human demonstrations. Specifically, we assess the impacts of trace optimization (row 2), exploration sampling (rows 3,4), and action sampling in response to perception hallucinations (row 6). Removing trace optimization decreases success rate by 0.02 while increasing simulation steps and the number of planning cycles. These results are intuitive; while human demonstrations do eventually lead to task success, the number of interaction steps is unnecessarily high due to sub-optimal action repetitions. This behavior is then learned and repeated by the planning model. Moving on, our skewed sampling of exploration actions teaches the planner than objects are likely to be found earlier during exploration due to the ability to observe the wider scene while exploring the current furniture (i.e., seeing an object on a neighboring chair while exploring the couch). Encoding this characteristic in training yields a planner that is 38% more efficient in simulation steps and 31% more efficient in planning cycles (row 1 vs 3). Ablating exploration sampling entirely yields an increase to success within standard error while increasing simulation steps by 50%. A takeaway of this ablation is that while the model can reason *effectively* about sequencing exploration actions, the model gained *efficiency* by being exposed to such sequencing in the training data. Finally, we ablate the sampling of actions in response to objects hallucinated by the perception system. We find that exposing the model to object hallucinations enables recovery behaviors; when this training exposure is ablated, Success and Percent Complete drop by 0.04 and 0.03, respectively.

| Row | Perception | Planning Method | Success Rate ↑ | Percent Complete ↑ | Simulation Steps ↓ | Planning Cycles ↓ | Extraneous Actions ↓ |
|-----|-----------|----------------|----------------|-------------------|-------------------|------------------|---------------------|
| 1 |  | HD (ours) | 0.63 ± 0.02 | **0.74** ± 0.01 | **2110** ± 80 | **21.6** ± 0.6 | 0.17 ± 0.01 |
| 2 | GT | w/o Trace Optimization | 0.61 ± 0.02 | 0.70 ± 0.01 | 2760 ± 90 | 29.1 ± 0.7 | 0.14 ± 0.01 |
| 3 |  | w/ Uniform Explore Sampling | 0.63 ± 0.02 | 0.72 ± 0.01 | 2910 ± 90 | 28.4 ± 0.7 | **0.12** ± 0.01 |
| 4 |  | w/o Explore Sampling | **0.64** ± 0.02 | **0.74** ± 0.01 | 3160 ± 110 | 26.8 ± 0.7 | **0.12** ± 0.01 |
| 5 | Learned | HD (ours) | **0.55** ± 0.02 | **0.69** ± 0.01 | 3040 ± 110 | 30.2 ± 0.8 | 0.17 ± 0.01 |
| 6 |  | w/o Hallucination Sampling | 0.51 ± 0.02 | 0.66 ± 0.01 | **2480** ± 90 | **24.9** ± 0.6 | 0.27 ± 0.01 |

Table 7: **Ablations of Human Demonstration Training.** Given the trace derivation method presented in Section 4.2, we evaluate the effects of trace optimization (row 2), skewed sampling of exploration actions (row 3), exploration sampling in its entirety (row 4), and the sampling of actions that respond to perception hallucinations (row 6).

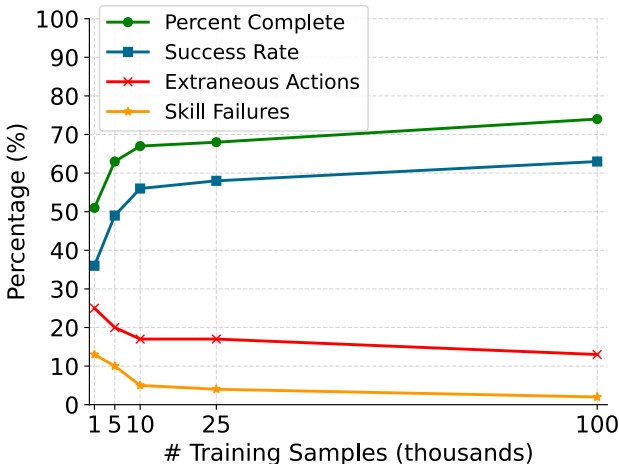

Figure 6: **Planning Performance Scales with Training Data.** Performance of our learning from demonstrations method on the PARTNR validation split with GT perception. Training data is scaled from 1,000 demonstrations to 100,000 demonstrations. This plot shows that with more training data, task success increases and inefficiencies (extraneous actions, skill failures) decrease.

### A.6.3 SCALING EXPERIMENTS

With Figure 6, we show that our method for deriving planning training data from human demonstrations yields increased performance with increased data. We demonstrate this in the perceptual setting of ground-truth EPM representations; scaling human demonstrations of PARTNR tasks from 1,000 to 100,000 tasks yields a performance increase by 0.27 success ($0.36 \rightarrow 0.63$) and decreases the rate of skill coordination failures by 0.11 ($0.13 \rightarrow 0.02$). Across our tested data scales (1k, 5k, 10k, 25k, 100k), we observe monotonic improvements to success.

### A.7 DETAILS AND BASELINES: DDAFT

*DDAFT* requires a set of tasks available for training $\tau \in \mathcal{T}$ and a task-conditioned initial planning policy $\pi_0(a|s,\tau)$. *DDAFT* first rolls out $\pi_0$ on all episodes in $\mathcal{E}$ producing an initial dataset of traces with task rewards $\{(x_0, r_0), \ldots (x_n, r_n)\} = \mathcal{D}_0$. The process continues by alternating between finetuning a planning model on the latest dataset

$$\theta_i = \arg\min_\theta \mathbb{E}_{x,r\sim D_{i-1}} \mathcal{L}(\theta, x, r), \qquad (1)$$

and sampling new traces to form a new dataset.

Instead of sampling new traces uniformly across the training episodes, *DDAFT* samples so that minimal exploration happens on episodes for which high-quality planning data already exists. Instead exploration is focused on episodes for which the dataset currently contains no successful examples.

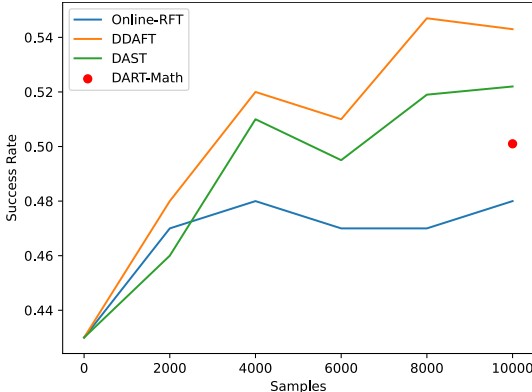

Figure 7: Success rate on the PARTNR val split using the learned EPM vs number of training samples on the PARTNR train-2k split.

We do this by sampling which episodes to generate new traces on from a distribution induced by the softmax over the failure rates of each episode in the dataset so far. We compute the failure rate per task $\tau_i$ across a dataset $\mathcal{D}$ as:

$$\boldsymbol{\tau}_i = \frac{\sum_{x,r \in D} \mathbb{1}_{x=\tau_i} \cap \mathbb{1}_{r=0}}{\sum_{x \in D} \mathbb{1}_{x=\tau_i}} \tag{2}$$

Where $r$ represents the terminal reward (1 or 0) for trace $x$ in the dataset, and $\mathbb{1}_{x=\tau_i}$ indicates that trace $x$ is an instance of episode $\tau_i$. Given this, the probability of selecting episode $\tau_i$ for exploration is

$$p(\tau_i) = \sigma_a(\boldsymbol{\tau})_i \tag{3}$$

where $\sigma_a$ represents the softmax operator with temperature $a$. An outline of the *DDAFT* algorithm can be found in Algorithm 1. Hyperparameters for *DDAFT* training can be found in Table 8. Training for each iteration took approximately 40 GPU/Hours on A100 cards, and collecting traces for each iteration took an average of 34.0 GPU*Hours for LLM inference (planning), and 62.5 GPU*Hours for VLM inference (EPM).

---

**Algorithm 1** The *DDAFT* algorithm

---

 1: **Input:** Set of episodes $\mathcal{T}$, initial policy $\pi_0(a|s,\tau)$
 2: $D_0 \leftarrow$ Rollouts from $\pi_0$ on all tasks in $\mathcal{T}$
 3: **for** $i \in \{1, 2, \dots n\}$ **do**
 4:     Fit $\pi_i$ to minimize loss on $\mathcal{D}_{i-1}$ according to Equation 1
 5:     Initialize $\mathcal{D}_i \leftarrow \mathcal{D}_{i-1}$
 6:     **for** $j \in \{1, 2, \dots m\}$ **do**
 7:         Sample a task $\tau_j \sim p(\tau)$ according to Equation 3
 8:         $x, r \leftarrow$ Rollout $\pi_i$ on task $\tau_j$
 9:         $\mathcal{D}_i \leftarrow \mathcal{D}_i \cup (x, r)$
10:     **end for**
11: **end for**

---

### A.7.1 ADDITIONAL RL EXPERIMENTS

To better understand the effect of the different components of *DDAFT* we compare against the following baselines:

- **DART-Math** Tong et al. (2024): DART-Math does not sample instructions based on a dynamic difficulty measure, but performs one initial difficulty estimate. It then samples a fixed set of traces based on a sampling frequency for each episode linearly proportional to its estimate difficulty. For this baseline we sample 10,000 traces from `Llama3.3-70b` according to DART difficulty curve.

Table 8: Hyperparameters for *DDAFT* experiments, shared across other baselines unless otherwise specified.

| Hyperparameter | Values |
|---|---|
| Learning Rate | $1e^{-5}$ |
| LR Schedule | Linear Decay with warmup |
| LoRA $\alpha$ | 64 |
| LoRA Rank | 128 |
| LoRA Dropout | 0.01 |
| LoRA Params | query and value projections |
| *DDAFT* Softmax $\alpha$ | 3 |
| Batch Size | 8 |
| Training Duration | 25 Epochs |
| *DDAFT* Iterations | 5 |
| Samples per *DDAFT* iteration | 2000 |

- **Online-RFT** Yuan et al. (2023): This is a variant of RFT which iteratively generates traces for training, and fitting a model on all successful traces. This the same as DDAFT except instructions are sampled uniformly from the training instructions.

- **DAST** Xue et al. (2025): DAST is a concurrent work which also propose dynamic difficulty based sampling. Instead of shaping the sampling distribution with a softmax over failure-rates, DAST utilizes defines a hard-coded mapping of failure-rate to sampling frequency. This means it does not adjust difficulty estimations within one sampling iteration.

Since DART-Math and DAST both require a pre-computed difficulty score before sampling, both of those methods have initial difficulty scores computed using 10,000 samples from `Llama3.3-70b` before online sampling starts. For DART-Math, all of the 10,000 samples for training were collected using the stronger model (Llama3.3) while the online methods all use samples from the current working model (finetuned from `LLama3.1-8b`).

Results using these baselines combined with learned EPM perception can be found in Figure 7. We see that online training with dynamic difficulty estimation out perform static difficulty estimation (DART-Math) or online training with no difficulty aware sampling (Online-RFT). We see that DDAFT outperforms a concurrently developed method which also explores dynamic difficulty estimation (DAST). Since DAST relies on a hardcoded set of sampling frequencies based on failure rates calculated before each sampling iteration, it does not adjust sampling rates within a single iteration. While we see greater sample efficiency with *DDAFT* as compared to DART, both methods will likely converge to the same performance as the number of samples approaches infinite.

## A.8 EXTENDED PLANNING RESULTS

We present extended evaluation metrics for all planning models in Table 9. We include results of models broken down by task type in PARTNR to evaluate the implications of perception systems on planning behavior. We also include new metrics for skill coordination to assess the implications of perception systems on ability to call and coordinate skills. Namely, we include *Skill Success*, the percentage of skill calls that are successful, *Episodes Without Skill Failure*, the number of episodes that include zero failed skill calls, and *Success After Skill Failure*, the Success Rate of the subset of episodes that involved at least on skill failure. We highlight some insights from these results below.

**Learned Perception Exacerbates Sensitivity to Task Type.** Our models have higher performance on Constraint-Free and Spatial tasks than Temporal tasks. This difference is exacerbated by learned perception; while Temporal performance of `HD` drops 0.13 success from Constraint-Free with perfect perception (row 14 vs 16), the same setting drops 0.20 success with learned perception (54% more; row 34 vs 36). In this case, we observe failures of the planner to manage task complexity while mitigating limitations in perception. This showcases the importance of jointly evaluating planners and perception systems on the most challenging tasks.

| Row | Method | Perception | Task Type | Success Rate ↑ | Percent Complete ↑ | Simulation Steps ↓ | Planning Cycles ↓ | Extraneous Actions ↓ | Skill Success ↑ | Episodes W/o Skill Failure ↑ | Success After Skill Failure ↑ |
|---|---|---|---|---|---|---|---|---|---|---|---|
| 1 | DynaMem | GT | All | 0.07 ± 0.01 | 0.15 ± 0.01 | 5520 ± 130 | - | - | - | - | - |
| 2 | | | Constraint-Free | 0.08 ± 0.02 | 0.16 ± 0.02 | 4790 ± 170 | - | - | - | - | - |
| 3 | | | Spatial | 0.10 ± 0.02 | 0.18 ± 0.02 | 4490 ± 150 | - | - | - | - | - |
| 4 | | | Temporal | 0.03 ± 0.01 | 0.12 ± 0.02 | 7340 ± 280 | - | - | - | - | - |
| 5 | PP | GT | All | 0.51 ± 0.02 | 0.67 ± 0.01 | 1570 ± 40 | 17.55 ± 0.36 | 0.19 ± 0.01 | 0.93 ± 0.00 | 0.56 ± 0.02 | 0.41 ± 0.03 |
| 6 | | | Constraint-Free | 0.52 ± 0.03 | 0.70 ± 0.02 | 1450 ± 60 | 16.65 ± 0.54 | 0.16 ± 0.02 | 0.94 ± 0.00 | 0.56 ± 0.03 | 0.45 ± 0.05 |
| 7 | | | Spatial | 0.59 ± 0.03 | 0.72 ± 0.02 | 1350 ± 60 | 15.70 ± 0.64 | 0.14 ± 0.02 | 0.92 ± 0.00 | 0.61 ± 0.03 | 0.48 ± 0.05 |
| 8 | | | Temporal | 0.42 ± 0.03 | 0.57 ± 0.03 | 1930 ± 85 | 20.53 ± 0.67 | 0.27 ± 0.02 | 0.93 ± 0.00 | 0.50 ± 0.03 | 0.32 ± 0.04 |
| 9 | PP+DDAFT (ours) | GT | All | 0.66 ± 0.02 | 0.78 ± 0.01 | 2160 ± 60 | 22.29 ± 0.55 | 0.19 ± 0.01 | 0.96 ± 0.00 | 0.63 ± 0.02 | 0.54 ± 0.03 |
| 10 | | | Constraint-Free | 0.69 ± 0.03 | 0.80 ± 0.02 | 2130 ± 110 | 22.51 ± 0.96 | 0.18 ± 0.02 | 0.96 ± 0.00 | 0.61 ± 0.03 | 0.61 ± 0.05 |
| 11 | | | Spatial | 0.77 ± 0.03 | 0.85 ± 0.02 | 1680 ± 90 | 17.96 ± 0.89 | 0.13 ± 0.02 | 0.96 ± 0.00 | 0.70 ± 0.03 | 0.56 ± 0.06 |
| 12 | | | Temporal | 0.53 ± 0.03 | 0.68 ± 0.02 | 2660 ± 110 | 26.40 ± 0.92 | 0.26 ± 0.02 | 0.97 ± 0.00 | 0.57 ± 0.03 | 0.47 ± 0.05 |
| 13 | HD (ours) | GT | All | 0.63 ± 0.02 | 0.74 ± 0.01 | 2110 ± 80 | 21.6 ± 0.6 | 0.17 ± 0.01 | 0.98 ± 0.01 | 0.90 ± 0.01 | 0.31 ± 0.05 |
| 14 | | | Constraint-Free | 0.65 ± 0.03 | 0.73 ± 0.03 | 2700 ± 150 | 24.88 ± 0.97 | 0.11 ± 0.02 | 0.99 ± 0.00 | 0.92 ± 0.02 | 0.45 ± 0.11 |
| 15 | | | Spatial | 0.70 ± 0.03 | 0.77 ± 0.02 | 2130 ± 130 | 21.03 ± 0.92 | 0.09 ± 0.02 | 0.98 ± 0.00 | 0.89 ± 0.02 | 0.29 ± 0.09 |
| 16 | | | Temporal | 0.52 ± 0.03 | 0.64 ± 0.03 | 2750 ± 130 | 27.95 ± 0.92 | 0.20 ± 0.02 | 0.97 ± 0.00 | 0.87 ± 0.02 | 0.25 ± 0.08 |
| 17 | HD+DDAFT (ours) | GT | All | 0.68 ± 0.02 | 0.80 ± 0.01 | 2550 ± 90 | 22.71 ± 0.53 | 0.15 ± 0.01 | 0.97 ± 0.00 | 0.88 ± 0.01 | 0.27 ± 0.05 |
| 18 | | | Constraint-Free | 0.72 ± 0.03 | 0.84 ± 0.02 | 2560 ± 150 | 22.32 ± 0.91 | 0.11 ± 0.02 | 0.98 ± 0.00 | 0.90 ± 0.02 | 0.36 ± 0.04 |
| 19 | | | Spatial | 0.75 ± 0.03 | 0.83 ± 0.02 | 2120 ± 150 | 18.85 ± 0.88 | 0.12 ± 0.02 | 0.96 ± 0.00 | 0.87 ± 0.02 | 0.18 ± 0.07 |
| 20 | | | Temporal | 0.57 ± 0.03 | 0.73 ± 0.02 | 2960 ± 160 | 26.99 ± 0.89 | 0.21 ± 0.02 | 0.97 ± 0.00 | 0.88 ± 0.02 | 0.30 ± 0.09 |
| 21 | DynaMem | DynaMem | All | 0.03 ± 0.01 | 0.11 ± 0.01 | 5090 ± 120 | - | - | - | - | - |
| 22 | | | Constraint-Free | 0.02 ± 0.01 | 0.10 ± 0.01 | 4520 ± 180 | - | - | - | - | - |
| 23 | | | Spatial | 0.05 ± 0.01 | 0.14 ± 0.02 | 3970 ± 139 | - | - | - | - | - |
| 24 | | | Temporal | 0.01 ± 0.01 | 0.08 ± 0.01 | 6850 ± 270 | - | - | - | - | - |
| 25 | PP | Learned | All | 0.46 ± 0.02 | 0.65 ± 0.01 | 1850 ± 50 | 19.72 ± 0.40 | 0.28 ± 0.01 | 0.89 ± 0.00 | 0.43 ± 0.02 | 0.36 ± 0.04 |
| 26 | | | Constraint-Free | 0.51 ± 0.03 | 0.72 ± 0.02 | 1760 ± 80 | 18.50 ± 0.63 | 0.22 ± 0.02 | 0.91 ± 0.00 | 0.47 ± 0.03 | 0.46 ± 0.04 |
| 27 | | | Spatial | 0.56 ± 0.03 | 0.72 ± 0.02 | 1510 ± 62 | 16.40 ± 0.57 | 0.21 ± 0.02 | 0.90 ± 0.00 | 0.50 ± 0.03 | 0.45 ± 0.04 |
| 28 | | | Temporal | 0.30 ± 0.03 | 0.50 ± 0.02 | 2300 ± 92 | 24.41 ± 0.77 | 0.40 ± 0.02 | 0.87 ± 0.00 | 0.30 ± 0.03 | 0.22 ± 0.03 |
| 29 | PP+DDAFT (ours) | Learned | All | 0.58 ± 0.02 | 0.74 ± 0.01 | 2200 ± 60 | 23.24 ± 0.53 | 0.27 ± 0.01 | 0.90 ± 0.00 | 0.41 ± 0.02 | 0.46 ± 0.02 |
| 30 | | | Constraint-Free | 0.61 ± 0.03 | 0.77 ± 0.02 | 2170 ± 110 | 23.33 ± 0.92 | 0.26 ± 0.02 | 0.90 ± 0.00 | 0.40 ± 0.03 | 0.49 ± 0.04 |
| 31 | | | Spatial | 0.71 ± 0.03 | 0.82 ± 0.02 | 1660 ± 90 | 18.38 ± 0.80 | 0.21 ± 0.02 | 0.90 ± 0.00 | 0.46 ± 0.03 | 0.60 ± 0.04 |
| 32 | | | Temporal | 0.41 ± 0.03 | 0.62 ± 0.02 | 2790 ± 110 | 28.08 ± 0.92 | 0.34 ± 0.02 | 0.89 ± 0.00 | 0.37 ± 0.03 | 0.30 ± 0.04 |
| 33 | HD (ours) | Learned | All | 0.55 ± 0.02 | 0.69 ± 0.01 | 3040 ± 110 | 30.23 ± 0.76 | 0.17 ± 0.01 | 0.95 ± 0.00 | 0.75 ± 0.02 | 0.32 ± 0.03 |
| 34 | | | Constraint-Free | 0.59 ± 0.03 | 0.72 ± 0.02 | 3190 ± 190 | 30.82 ± 1.35 | 0.11 ± 0.02 | 0.98 ± 0.00 | 0.80 ± 0.03 | 0.40 ± 0.07 |
| 35 | | | Spatial | 0.66 ± 0.03 | 0.78 ± 0.02 | 2560 ± 170 | 26.09 ± 1.25 | 0.11 ± 0.02 | 0.95 ± 0.00 | 0.75 ± 0.03 | 0.50 ± 0.06 |
| 36 | | | Temporal | 0.39 ± 0.03 | 0.56 ± 0.03 | 3380 ± 190 | 33.93 ± 1.30 | 0.29 ± 0.03 | 0.94 ± 0.00 | 0.70 ± 0.03 | 0.11 ± 0.04 |
| 37 | HD+DDAFT (ours) | Learned | All | 0.58 ± 0.02 | 0.74 ± 0.01 | 2250 ± 70 | 23.62 ± 0.55 | 0.23 ± 0.01 | 0.91 ± 0.00 | 0.44 ± 0.02 | 0.45 ± 0.02 |
| 38 | | | Constraint-Free | 0.62 ± 0.03 | 0.77 ± 0.02 | 2125 ± 110 | 22.82 ± 0.90 | 0.20 ± 0.02 | 0.91 ± 0.00 | 0.43 ± 0.03 | 0.49 ± 0.04 |
| 39 | | | Spatial | 0.67 ± 0.03 | 0.79 ± 0.02 | 1775 ± 100 | 19.32 ± 0.87 | 0.19 ± 0.02 | 0.92 ± 0.00 | 0.51 ± 0.03 | 0.54 ± 0.04 |
| 40 | | | Temporal | 0.43 ± 0.03 | 0.65 ± 0.02 | 2870 ± 130 | 28.95 ± 0.98 | 0.30 ± 0.02 | 0.90 ± 0.00 | 0.38 ± 0.03 | 0.33 ± 0.04 |

Table 9: **Extended Planning Results** of Table 3, including a performance breakdown by task type in PARTNR (Constraint-Free, Spatial, and Temporal) along with additional skill coordination metrics.

**HD Models Are Strong at Coordinating Skills.** HD and HD-DDAFT have the highest Skill Success rates across both GT and EPM perception, demonstrating that data derived from human demonstrations teaches planning models to effectively coordinate skills. This is exemplified by the metric Episodes without Skill Failure, which shows that zero-shot models in the GT setting have just a 56% rate, compared with HD which has a 90% rate (row 5 vs 13). Further, DDAFT increases the overall success rate of planning models without negatively impacting skill coordination.

*DDAFT* **Enables Robustness to Skill Failures.** In episodes that include skill failures, *DDAFT* increases success of the PP model from 0.36 to 0.46 (row 25 vs 29) and increases success of the HD model from 0.32 to 0.45 (row 33 vs 37). We suspect that these improvements are brought about by exposing the planner to in-distribution skill failures during training.

**Runtime Analysis.** The inference time of perception and planning systems is an important consideration toward real-world applicability. We compare EPM to the DynaMem baseline in this regard. When running at the same frequency and on the same hardware (A100 GPU), EPM takes on average 0.44 seconds per update, where DynaMem takes 1.57 seconds per update (build and query). This is a 3.5x delta. When also considering planning efficiency, we observe that our method requires on average 5 minutes per episode, while DynaMem may take up to 6 hours in larger scenes. Overall, this shows that time consumption is not a major concern of our method. Further, we note while EPM takes <1MB of memory to keep text representation, DynaMem uses  4GB of memory to keep the entire featurized point-cloud for querying.

**Note on Implementation Differences With PARTNR.** PARTNR Chang et al. (2025) presents single-agent ReAct achieving a 0.73 success rate, yet our similar PP baseline with ground-truth perception achieves lower success (0.51; row 5). The primary distinction lies in the perception model. In PARTNR, the planner used oracle perception where if even one pixel of an object was seen at any distance, the semantic class and 3D position is provided to the planner using simulator information. In EPM experiments with ground truth perception (rows 1-20 and Table 3 rows 1-5), thresholds were applied to better approximate realistic perception capabilities. Objects were only considered detected

if they occupied at least 0.1% of the viewing area and were between 0.2 and 6 meters from the agent. Zero-Shot results jump to the levels reported in PARTNR when these thresholds are removed.

**Planner performance by episode length** To investigate the effect of episode length on planning performance we present the results of HD trained react style planners broken down by episode length (Table 10). We can see for all methods, longer episodes are more challenging in general. However the gap in performance between learned EPM perception and GT is largely due to medium and long episode performance, indicating that compounding perception errors remain an issue. We find that training with DDAFT improves performance in these more challenging episodes by teaching the planner to be robust to perception errors. See A.9 for a qualitative example.

| Method | Perception | Short Episode SR | Medium Episode SR | Long Episode SR |
|--------|-----------|------------------|-------------------|-----------------|
| HD | GT | 0.82 | 0.74 | 0.26 |
| HD+DDAFT | GT | 0.84 | 0.82 | 0.31 |
| HD | Learned | 0.82 | 0.50 | 0.19 |
| HD+DDAFT | Learned | 0.82 | 0.64 | 0.25 |

Table 10: Task planning performance of models trained on human demonstrations (HD) and human demonstrations + policy improvement (DDAFT) broken down by episode length. Short episodes are episodes which ran for fewer than 13 skill calls. Medium episodes were episodes which ran for between 13 and 16 skill calls. Long episodes were episodes which ran longer than 26 skill calls. Total skill calls were capped at 50.

## A.9 QUALITATIVE RESULTS

Below is an example of the *DDAFT* trained model planning around perception mistakes on novel tasks. In this task the agent must find a laptop and move it to the couch. We see that the perception system mistakenly detects a laptop early in the episode. The agent tries to pick this but fails (because there is actually no object at the detected location). Instead of trying again with the hallucinated object it continues exploring the scene until another laptop is detected. Eventually we see that this false positive from earlier in the episode is corrected by EPM and removed from the representation. Some pieces of the full prompt have been omitted for brevity. See section A.4 for details on the full prompt.

---

**Planning Around Detection Failures**

```
Task: Move the book and laptop from the bedroom chest of drawers to the living room
couch. Place them next to each other on the couch.

This is the list of objects seen so far, together with their coordinates:
...
Object.116: couch in living_room_1
...
Object.144: chest of drawers in bedroom_1
Object.145: chest of drawers in bedroom_1
Object.146: chest of drawers in bedroom_3
Object.147: chest of drawers in bedroom_3
Object.148: chest of drawers in bedroom_1

...

Navigate[Object.144]
Assigned!<|eot_id|><|start_header_id|>user<|end_header_id|>

Result: Successful execution!
The following objects were updated:
Add Object.153: candle with the Object.137
Add Object.154: cushion with the Object.121
Add Object.155: vase with the Object.144
Add Object.156: helmet with the Object.144
<|eot_id|><|start_header_id|>assistant<|end_header_id|>

ExploreFurniture[Object.144]
Assigned!<|eot_id|><|start_header_id|>user<|end_header_id|>

Result: Successful execution!
No object updates.
<|eot_id|><|start_header_id|>assistant<|end_header_id|>
Navigate[Object.146]
Assigned!<|eot_id|><|start_header_id|>user<|end_header_id|>

Result: Successful execution!
The following objects were updated:
Add Object.157: laptop with the Object.145
Add Object.158: cushion with the Object.127
Add Object.159: box with the Object.107
<|eot_id|><|start_header_id|>assistant<|end_header_id|>

Pick[Object.157]
Assigned!<|eot_id|><|start_header_id|>user<|end_header_id|>

Result: Unexpected failure! - No object found near the target position
-10.924965127447484, 0.5990211561596542, -9.398278611277211
No object updates.
<|eot_id|><|start_header_id|>assistant<|end_header_id|>

Navigate[Object.145]
Assigned!<|eot_id|><|start_header_id|>user<|end_header_id|>

Result: Successful execution!
The following objects were updated:
Add Object.160: box with the Object.128
Add Object.161: laptop stand with the Object.145
Add Object.162: multiport stand with the Object.145
Add Object.163: laptop with the Object.145
Add Object.164: vase with the Object.145
<|eot_id|><|start_header_id|>assistant<|end_header_id|>
```

```
Pick[Object.163]
Assigned!<|eot_id|><|start_header_id|>user<|end_header_id|>
Result: Successful execution!
The following objects were updated:
Update Object.163 (picked): laptop
<|eot_id|><|start_header_id|>assistant<|end_header_id|>

Navigate[Object.116]
Assigned!<|eot_id|><|start_header_id|>user<|end_header_id|>

Result: Successful execution!
The following objects were updated:
Add Object.165: cushion with the Object.103
<|eot_id|><|start_header_id|>assistant<|end_header_id|>

Place[Object.163, on, Object.116, None, None]
Assigned!<|eot_id|><|start_header_id|>user<|end_header_id|>

Result: Successful execution!
The following objects were updated:
Update Object.163: laptop with the Object.116
<|eot_id|><|start_header_id|>assistant<|end_header_id|>
...
Navigate[Object.144]
Assigned!<|eot_id|><|start_header_id|>user<|end_header_id|>
Result: Successful execution!
The following objects were updated:
Remove Object.157
<|eot_id|><|start_header_id|>assistant<|end_header_id|>
...
```

## A.10   REAL-WORLD EXPERIMENTS WITH THE EPM

We present a qualitative analysis of planning with the EPM in the real world. For these experiments, we select the subset of objects that were detected consistently and correctly in our offline real-world experiments. We do this so that we can focus on efficacy of the EPM in planning in these experiments. We evaluated four instructions by running five runs for each. Active objects are randomly placed on the indicated receptacle each run. We use `PP-DDAFT` planner for these experiments with the learned EPM.

**System.** Our experiments are based on the systems presented in Yokoyama et al. (2023). We extend this system with a hand-annotated house layout that is used as $M_{fur}$ input by the EPM and our planner. This house layout is also used by the navigation primitive in the real-world to approach a given receptacle. In particular, note that this system uses OWL-ViT 2 to detect the object tasked with picking and uses the image pixel on the object to then command the Boston Dynamics' grasp API on Spot. Thus there is an additional interaction here between complex multi-modal models that is different from our simulation setup. This real-world system also includes a retry mechanism which keeps re-calling actions if the gripper is not "empty" after rolling-out place policy or reports being "empty" after rolling-out pick policy.

**Instructions.** We use the following scenarios for our real world experiments:

- Scenario 1
  `Instruction:` Move the cup from coffee table in living room to the sink in kitchen.
  `Scene initialization:` Coffee table - cup, book, decor item. Sink - empty.
- Scenario 2
  `Instruction.` Move plush toy from sofa to the bed and cup from bed to the coffee table.
  `Scene Initialization.` Sofa - cushions, pineapple toy. Bed - cup.
- Scenario 3
  `Instruction.` We need to clean up the office. Move the cup and the bottle from office desk to the dining table.
  `Scene Initialization.` Desk - cup, bottle, backpack. Dining table - bowl.
- Scenario 4
  `Instruction.` Bring the bottle from the counter to the coffee table and move the toy from the bed to the couch.

`Scene Initialization`. Kitchen counter - bottle. Bed - plush toy. Sofa - cushion.

**Results.** We provide two metrics for these experiments: planning success and execution success. For the former metric, we consider all episodes where the EPM and planner predicted the correct plan to achieve the instruction successfully even if skills failed. For execution success, we only consider those instructions which change the environment to the goal state. We see 70% planning success (14/20) and 55% execution success (11/20).

**Discussion.** The EPM not adding active objects (those mentioned in the task) is the most common failure mode, followed by adding objects as a different category altogether. This also leads to suboptimal behaviors from the planner, for example we see planner calling pick on alternate object detected on the identified receptacle in the absence of the intended one. We also note absence of exploratory behavior, e.g. navigating to the active receptacle does not result in the active object being added to the memory, and we do not see the planner call "explore" on the receptacle either. This motivates further work in studying the co-emergence of planning and perception together.

### A.11 SOCIETAL IMPACTS

We include a few societal considerations of our work.

**Labor Impact.** The systems we develop in this work amount to a step toward the automation of household tasks. There are both benefits and risks that arise from this goal; a benefit is scalable, efficient, and cheap augmentation of labor to serve individuals or families that have restrictions on time or physical ability. Some risks include the undercutting of the human labor force.

**Bias.** Training of the EPM and planner elicits behaviors representative of the training domains. While the HSSD environments in PARTNR are diverse in size and scale, they contain clean, minimally-cluttered spaces that reflect a particular collection of real-world spaces. Thus, the models we train in this paper will have a bias towards effective operation in these settings.

### A.12 USE OF LARGE LANGUAGE MODELS

We used Large Language Models to polish some parts of the text (fixing grammar issues, shorten sentences, etc). Below, we detail how we use LLMs in our work, and which models we use in each case.

**High-Level Planning**: We use Llama-3.3-70b-Instruct to generate plans for the PARTNR-Pretrained (Chang et al., 2025) planning baseline. The model is prompted zero-shot to generate high-level actions based on a text description of the task and current state of the environment. We also use Llama-3.1-8b-Instruct as the backbone for our method, which we fine-tune on traces from PARTNR-Prertained, as well as the human demonstration dataset, described in Sec. 4.2.

**Human-In-The-Loop Data Generation**: As described in Sec. 4.2, we use CodeLlama-70b to derive exploration targets from the Human-In-The-Loop data. Given a description of the scene and task, the LLM generates a set of furniture candidates, which are used to generate exploration targets.

### A.13 LICENSES FOR EXISTING ASSETS

**HSSD.** The Habitat Synthetic Scenes Dataset (HSSD Khanna et al. (2024)) is licensed under CC BY-NC 4.0 and was downloaded from `https://huggingface.co/datasets/hssd/hssd-hab` (accessed May 22, 2025).

**PARTNR.** The PARTNR Chang et al. (2025) code is licensed under MIT and was accessed from `https://github.com/facebookresearch/partnr-planner` (accessed May 22, 2025). The PARTNR dataset is licensed under CC BY-NC and was downloaded from `https://huggingface.co/datasets/ai-habitat/partnr_episodes` (accessed May 22, 2025).

**Llama 3.** The Llama 3 models used in this work Grattafiori et al. (2024) are licensed under the META LLAMA 3 COMMUNITY LICENSE AGREEMENT and were downloaded from `https://huggingface.co/meta-llama` (accessed May 22, 2025).

**Llava OneVision.** The `Llava-Onevision-7b-Qwen2` model Li et al. (2024) is licensed under Apache 2.0 and was downloaded from `https://huggingface.co/lmms-lab/llava-onevision-qwen2-7b-ov` (accessed May 22, 2025).

**DynaMem.** Our DynaMem baseline model Liu et al. (2024a) is adapted from `https://github.com/hello-robot/stretch_ai/blob/main/docs/dynamem.md` (accessed May 22, 2025) which was released with the MIT license.