# OpenReview forum: "Planning with an Embodied Learnable Memory"
_ICLR.cc/2026/Conference — ICLR 2026 Poster_

### Official Review · Reviewer_esNZ · 2025-10-30

**Soundness:** 3
**Presentation:** 3
**Contribution:** 3
**Rating:** 6
**Confidence:** 3

**Summary:**

This paper presents Embodied Perception Memory (EPM), a learnable memory representation for long-horizon embodied planning in dynamic indoor environments. This representation enables end-to-end scene information operations (addition, removal, update), adopts object-centric encoding (3D coordinates + textual state/context descriptions), and eliminates explicit task-based queries by generating LLM-compatible textual scene representations. In addition, a two-stage tranining recipe is proposed to finetune LLM planners to leverage the EPM: first train the model on human demostration data to improve exploration and task execution, then leverage a novel online RL-based approach, named Dynamic Difficulty-Aware Fine-Tuning (DDAFT), to optimize planning via online sampling of difficult instructions.

**Strengths:**

1. The writing is clear and logically coherent, ensuring easy readability for readers.
2. The proposed memory representation is well-suited for tracking object states in dynamic indoor environments, and it exhibits several key advantages:
 (1) Easy maintenance: Unlike complex scene graphs that detail the relative relationships between all objects, EPM adopts a simpler spatial representation—one that focuses only on the relationships between dynamic objects and infrequently movable furniture.
 (2) Lightweight design: Compared with explicit semantic mapping techniques, EPM eliminates the need to perform segmentation, open-vocabulary encoding, and dynamic map updates.
3. The performance results are robust, particularly when planners are paired with the proposed memory under learned perception settings.

**Weaknesses:**

1. In Section 4.2, the use of human demonstration data for training is introduced; however, the description of how to acquire human demonstration data based on tasks and memory lacks clarity.
2. How is the instance association problem in the perception process addressed? It appears that the same instance may be recorded as multiple instances due to perception and localization errors. When applied to real world, has consideration been given to adding an operation type for merging entities?
3. In the inference phase, how is the information about which specific room each piece of furniture is located in (stored in memory) acquired? In the planning example in Appendix A.4, the descriptive information of the furniture is inconsistent with the content described in the Entity List and Embodied Perception Memory Input in Appendix A.2.

**Questions:**

See weakness above.

---

> ### Author Response · Authors · 2025-11-17
>
> We thank the reviewer for the insightful comments. We are glad that they valued simplicity and easy maintenance of our design, and the strength of our memory and planning system, as shown in our evaluation results. We address the reviewer’s concerns below.
>
>
> **In Section 4.2, the use of human demonstration data for training is introduced; however, the description of how to acquire human demonstration data based on tasks and memory lacks clarity.**
>
> The human demonstration data comes from the PARTNR paper where each PARTNR task was performed by a human in single-agent fashion in simulation. To learn a planner from these demonstrations while maintaining modularity of perception and planning, we describe a process of running the learned perception system while rolling out the human demonstration in Section 4.2. When the human plan is incompatible with the EPM state, we infer corrective planning actions and update the state of the EPM accordingly. We expand on the complete process in Appendix A.6.
>
> **How is the instance association problem in the perception process addressed? It appears that the same instance may be recorded as multiple instances due to perception and localization errors. When applied to real world, has consideration been given to adding an operation type for merging entities?**
>
> This is a great question, which highly influenced the design of the EPM. The EPM stores for every object its 3D location, which allows the EPM to implicitly determine if an observed object exists in the memory or not. However, localization errors could result in objects being added multiple times. To address this, the EPM uses the “Remove” operation which deletes previously mislocated or duplicated objects and adds new ones if necessary. To aid in this process, EPM is provided the distance from which the object was previously observed, which serves as a proxy for confidence (note that confidence calibration in VLMs is still an open problem [1]). We provide more details about how the EPM learns to remove objects in Appendix A.2.3.
>
>
> **In the inference phase, how is the information about which specific room each piece of furniture is located in (stored in memory) acquired? In the planning example in Appendix A.4, the descriptive information of the furniture is inconsistent with the content described in the Entity List and Embodied Perception Memory Input in Appendix A.2.**
>
> The example from A.4 shows the prompting for the planner and shows the initial furniture known to the planner at the beginning of the episode. The example in A.2 shows an example prompt for the embodied perception model. This may contain the initial furniture but also all objects which have been observed throughout the episode. The furniture lists are different because these two examples are from different episodes.

---

### Official Review · Reviewer_JRZU · 2025-11-01

**Soundness:** 2
**Presentation:** 3
**Contribution:** 2
**Rating:** 4
**Confidence:** 3

**Summary:**

This paper proposes an embodied planning framework that introduces a *dynamically learnable memory* to assist high-level task planning. The key idea is to maintain a textual memory that records the positions and states of currently observed objects, which is then injected into the input context of a large language model (LLM) planner. The planner conditions on this memory to produce high-level action decisions such as *Navigate*, *Pick*, and *Place*, allowing the system to reason about dynamic environments through language-conditioned planning.

**Strengths:**

- Incorporating a memory mechanism into embodied planning is conceptually valuable. It aligns with the broader goal of enhancing situational awareness and long-horizon reasoning in embodied agents.
- The proposed pipeline is relatively complete, combining human demonstration data and DDAFT to improve the planning capability of the LLM.
- The authors provide detailed implementation descriptions, which improve the reproducibility of the proposed approach.

**Weaknesses:**

The overall method is straightforward and lacks sufficient evidence to justify the necessity of the proposed memory mechanism. Essentially, the “memory” represents the textual description of observed object positions, which is appended as additional prompt context for the LLM planner to make decisions. This design raises questions about its true effectiveness: a Vision-Language Model (VLM) could directly perceive and infer object locations from images without requiring a separate process to extract and feed this information in textual form.

In the experiments, the authors compare models of different scales (7B and 70B) and their fine-tuned variants. The results show that supervised fine-tuning and DDAFT (a DPO-like reward fine-tuning method) indeed enhance model performance, but such improvement mainly reflects the benefit of additional data and optimization rather than the specific contribution of the memory mechanism. From the presented evidence, it remains unclear why this kind of dynamic memory is indispensable for embodied task planning.

Moreover, the evaluation is limited to the PARTNR benchmark. While this environment is a reasonable testbed, the empirical validation would be more convincing if the authors could extend experiments to other widely used embodied task planning benchmarks such as **ALFWorld** or **EmbodiedBench**. Including results from multiple environments would help demonstrate the generality and robustness of the proposed approach.

**Questions:**

1. **Why use an LLM planner instead of a VLM?**
   Since the memory content is purely a textual description of visual states, a Vision-Language Model could directly access equivalent spatial information from images. What is the rationale for choosing a text-only LLM over a multimodal model that already encodes spatial and visual grounding?
2. **Can you compare LLM w/ memory vs. VLM w/o memory vs. VLM w/ memory for the task planning?**
   Such a comparison would more convincingly isolate the contribution of the memory mechanism and clarify whether the observed gains come from the memory design
3. **Can you include additional benchmarks?**
   Evaluating on environments like **ALFWorld** or **EmbodiedBench** would strengthen the claim that the proposed approach generalizes across different embodied planning settings.

---

> ### Author Response · Authors · 2025-11-17
>
> We thank the reviewer for the valuable feedback. We appreciate that the reviewer valued the importance of incorporating memory into embodied planning, the completeness of our pipeline and the clarity of our paper. We address the reviewer’s concerns below.
>
>
> **Evidence of the proposed memory mechanism. The design raises questions about its true effectiveness: a Vision-Language Model (VLM) could directly perceive and infer object locations from images without requiring a separate process to extract and feed this information in textual form.**
>
> The VLM in our work does in fact perceive and infer object locations (the Add and Update operations predict pixel coordinates which are used to extract the 3D coordinates). We use the operators defined in Sec 3.2 to build a global state of the environment, which cannot be inferred from a single egocentric observation. To enable a VLM to directly predict the state of the environment, it should have access to all the history of visual observations, but currently there is no VLM that can handle such large number of frames, due to the computational requirements of the attention mechanism, which is why we use text to encode the previous information.
>
>
> **Improvement mainly reflects the benefit of additional data and optimization rather than the specific contribution of the memory mechanism.**
>
> The relative benefit of EPM vs a modular approach can be seen in comparing our method against DynaMem (Table 3 row 7 vs row 6). Even when using a pre-trained LLM with no additional finetuning, our method surpasses DynaMem on PARTNR’s long and complex tasks because the text representation produced by EPM is well suited to LLM planning. We see this same trend in real-world experiments as well (Supplementary A.10). Improvements due to DDAFT certainly are due to additional data as DDAFT is an RL process, however we would note that DDAFT is more sample efficient than other competitive RL methods in this setting (i.e. Online-RFT or DAST). We included comparisons against these methods in supplementary section A.7.1.
>
>
> **Why use an LLM planner instead of a VLM? Comparing with VLM baselines.**
>
> As described above, using an intermediate text representation to encode memory is necessary, since VLMs do not have enough context to encode all the agent’s history of observations. One alternative could be to use a single VLM to build the memory and plan with such memory. The reason why this approach is not effective is: 1) The VLM needs to be trained to perform planning and detection, reducing performance in both tasks and 2) It has been previously shown [1,2] that VLMs are significantly worse at reasoning and planning than LLMs, being unable to decompose the task into skills. The PARTNR paper, which proposes the benchmark our method evaluates on, analyzes the effectiveness of VLMs for planning, showing there is still a significant gap compared to LLM based methods. (https://openreview.net/forum?id=T5QLRRHyL1&noteId=jam218jjQ9).
>
>
> **Can you include additional benchmarks?**
>
> We note that we currently include experiments in two different environments in our paper, i.e. in multiple Habitat scenes and in a real-world apartment setting using BD Spot robot. For the real world, we include an in-depth analysis of perception failures (section 5.1.1) and a qualitative analysis of performance when deploying perception and planner models together (section A.10). Also, PARTNR is a comprehensive benchmark with 100,000 tasks in 200 scenes and 18,000 object instances. So, our results demonstrate that the proposed method generalizes across both large-scale simulated environments and a real-world robotic platform.
>
>
> *[1] Neau, Maëlic, et al. "GraSP-VLA: Graph-based Symbolic Action Representation for Long-Horizon Planning with VLA Policies." arXiv preprint arXiv:2511.04357 (2025).*
>
> *[2] Chen, Delong, et al. "Planning with reasoning using vision language world model." arXiv preprint arXiv:2509.02722 (2025).*

---

### Official Review · Reviewer_U9Bz · 2025-11-01

**Soundness:** 2
**Presentation:** 3
**Contribution:** 2
**Rating:** 6
**Confidence:** 3

**Summary:**

This paper proposes the Embodied Perception Memory (EPM), a textual list of entities (objects and furniture) and their relationships that uses a Vision-Language Model (VLM) to maintain and update. For planning, the authors employ a two-stage training strategy: (1) Imitation learning on planning trajectorirs derived from human demonstrations, which are post-processed to incorporate perception-specific exploration and robustness to memory errors; (2) A reinforcement fine-tuning method named Dynamic Difficulty-Aware Fine-Tuning (DDAFT), which adapts the DART-Math idea to the planning task during online policy improvement. The primary experimental validation on embodied planning is conducted on the PARTNR benchmark.

**Strengths:**

- The EPM provides a unified and interpretable text-based memory that seamlessly integrates with LLM planners, avoiding the need for complex querying mechanisms used in some prior works (e.g., feature matching in point clouds).

- The proposed DDAFT method effectively improves planning performance by dynamically focusing on difficult tasks during training.

**Weaknesses:**

- The idea of using a unified VLM to convert visual observations into a textual memory for planning has been explored in prior works [1-3].
- The experiments are conducted only on the PARTNR validation set. It is unclear how the method would perform on the test set, which is important for assessing generalization and fairness in benchmarking.
- The training data for EPM relies on simulator-privileged information (e.g., ground-truth object states and poses), which limits the scalability and applicability of the method to real-world settings where such information is unavailable.

[1] Context-aware planning and environment-aware memory for instruction following embodied agents, ICCV 2023
[2] STMA: A Spatio-Temporal Memory Agent for Long-Horizon Embodied Task Planning, arxiv 2025.02
[3] KARMA: Augmenting Embodied AI Agents with Long-and-short Term Memory Systems, ICRA 2025

**Questions:**

1. How does the system handle multi-object disambiguation? For example, if there are four apples and five milk bottles in the environment, how does EPM distinguish and track each instance in its textual representation?
2. For the Spot-Indoor dataset, are the annotations for adding/removing objects manually labeled, or are they constructed using the method described in Section 4.2 (i.e., via simulation replay and heuristics)?

---

> ### Author Response · Authors · 2025-11-17
>
> We thank the reviewer for the insightful comments. We are glad the reviewer valued the seamless integration of the memory with LLM planners and the effectiveness of DDAFT. We address the reviewer’s concerns below.
>
>
> **The idea of using a unified VLM to convert visual observations into a textual memory for planning has been explored in prior works [1-3].**
>
> We thank the reviewer for the pointers. While related, these works significantly differ from our approach, or assume privileged information. We will discuss these references in the paper.
>
> - [1] is a modular system that requires semantic segmentation, object detection, tracking and reassociation, along with a set of hand-crafted rules for the memory. Similar to our DynaMem baseline, the multi-stage approach significantly increases computation time. Furthermore it locates task relevant objects but does not include information about where they are in the scene. As such, it cannot handle queries such as: pick the knives that are on the counter. Also, because the memory is not learnable, it lacks error recovery mechanisms beyond its predefined rules.
> - [2] Assumes access to a text description of the observation. The approach is validated in a text-world environment, but it is unclear how it generalizes without privileged observations. Furthermore, it requires LLM calls to derive spatial relationships, aggregate those relationships and use them for planning, significantly increasing computational requirements.
> - [3] also proposes a modular approach similar to DynaMem, and requires privileged information. It works by building a 3D scene graph from the scene, using privileged information from a simulator which contains object names, properties and relationships. While the paper mentions the approach could work in the real world by using object detectors, it is unclear how to obtain object properties and relationships without privileged information. Our approach uses privileged information at training time, but only requires RGB and Depth during deployment.
>
>
> **The experiments are conducted only on the PARTNR validation set. It is unclear how the method would perform on the test set, which is important for assessing generalization and fairness in benchmarking.**
>
> The PARTNR test set is not publicly available so we cannot provide results on the test set.
>
>
> **The training data for EPM relies on simulator-privileged information (e.g., ground-truth object states and poses), which limits the scalability and applicability of the method to real-world settings where such information is unavailable.**
>
> We want to first emphasize that our method requires these annotations **only during training**, not at deployment or test time. In fact, we show in A.10 that we can deploy our system in the real world with a 70% planning success. While constructing a training dataset is easier in simulation, this privileged information can actually be derived from annotations in real-world datasets. For instance, Epic-HD [1] contains language annotations of objects together with their 3D location and human interaction labels. Using this data, it is possible to follow the procedure in A.2.3. to construct a dataset from real world data. See response “Improving the EPM with real-world data” to reviewer ACax for an example of using an EPM model with such data.
>
>
> **How does the system handle multi-object disambiguation?**
>
> For every object detected, the EPM tracks its 3D location in the environment. This information is used to ground and disambiguate objects. If an apple is seen in a location not stored in the EPM, it will be added into the memory. Otherwise, the previously stored object will be updated. If the agent is observing an empty area whereas EPM contains an object in that area, the EPM will issue a Remove command that will update the memory with the latest observation.
>
>
> **For the Spot-Indoor dataset, are the annotations for adding/removing objects manually labeled, or are they constructed using the method described in Section 4.2.**
>
> We manually annotated the objects in the Spot-Indoor dataset, listing the first frame where a given object is seen, and the frames at which an object moves. Given that we do not train the EPM with that dataset, we do not need to label any EPM operation in the dataset. Instead we run the EPM trained in simulation and measure how accurate it is at tracking the manually annotated objects. We will clarify in the revision.

---

### Official Review · Reviewer_ACax · 2025-11-03

**Soundness:** 3
**Presentation:** 3
**Contribution:** 2
**Rating:** 4
**Confidence:** 4

**Summary:**

This paper introduces Embodied Perception Memory (EPM), a unified vision–language model based memory system for embodied planning that represents the environment as a textual, object-centric world model updated from egocentric observations. EPM dynamically tracks objects and changes in long-horizon and enables seamless integration with LLM-based planners. To train planners to effectively leverage this representation, the authors propose two complementary methods: (i) imitation learning from human demonstration trajectories and (ii) Dynamic Difficulty-Aware Fine-Tuning (DDAFT), a value-function-free reinforcement learning algorithm that prioritizes training on harder tasks by sampling episodes based on failure rates. They evaluate EPM both as a perception module in isolation and as part of full embodied planning tasks. In particular, EPM outperforms existing memory architectures such as DynaMem and also surpasses LLM-based planners trained solely on the PARTNR dataset.

**Strengths:**

1. The writing is generally clear, with strong motivation and context provided for the problem setting.

2. The paper introduces a novel memory mechanism for embodied planning which is a conceptual advance over prior approaches that rely on separate perception modules, graph-based representations, or retrieval interfaces. The integration of a single VLM that simultaneously maintains, updates and exposes a textual world representation represents a meaningful rethinking of memory design for embodied agents.

3. The proposed approach tackles an increasingly important problem in embodied A, long-horizon planning in dynamic environments. The ability to maintain and evolve a textual world model from egocentric observations, while integrating seamlessly with language-based planners, has direct implications for future real-world embodied AI systems.

4. By unifying perception and memory within a single model, the approach reduces reliance on heavy symbolic querying or multi-module pipelines, which can be brittle and computationally expensive. This end-to-end design offers a promising direction toward scalable, real-time embodied planning systems.

5. The paper evaluates both perception and planning. The perception system is tested on both simulaton and real-world egocentric data, demonstrating promising transfer beyond simulation. Planning performance is assessed in challenging simulated environments, where EPM outperforms prior memory architectures (e.g., DynaMem) and LLM planners trained solely on PARTNR.

**Weaknesses:**

1. Real-world perception results do not show clear advantage over DynaMem: While the perception module is evaluated on real-world egocentric data, the reported results do not show a clear performance improvement over DynaMem in this setting. The authors attribute this to out-of-distribution objects that appear in real environments but are absent in simulation training. However, this highlights a practical limitation of the current approach: the system’s ability to generalize in open-world settings remains uncertain. It would be valuable to explore whether fine-tuning EPM on diverse real-world embodied datasets could reduce this gap. For example, large-scale egocentric datasets such as Ego4D, BEHAVIOR-1K real-world captures, or RT-1/RT-2 robot interaction datasets contain a wide variety of real objects and household interactions that could support domain adaptation.

2. No real-world validation of the planning pipeline: While the perception module is evaluated on real-world egocentric data, the planning results are restricted to simulation. Consequently, it remains uncertain how well the complete EPM-based planning pipeline would transfer to physical robots, where challenges such as sensor noise, actuation inaccuracies, and occlusions are more pronounced. Even small-scale real-robot experiments, or analysis demonstrating robustness to real-world observation and control noise, would significantly strengthen the claim that the method scales beyond simulation.

3. Incomplete comparison to recent memory systems and lack of computational analysis: While the paper compares against DynaMem, the evaluation omits other recent memory representations that also maintain object-centric world models, such as ConceptFusion, ConceptGraph etc. which would provide a more complete picture of performance relative to the state of the art in embodied memory systems. Additionally, although success rates and perception F1 scores are reported, the paper does not quantify computational efficiency or update overhead, which is central to the claimed scalability benefits. Reporting metrics such as time per memory update, insertion cost for new objects, query latency, and overall inference load per planning step would clarify the practical efficiency gains of EPM over modular graph-based or retrieval-based memory architectures.

4. Limitations of a text-only memory representation: A purely textual memory, while clean and LLM-friendly, may fail to capture fine-grained perceptual details needed for precise manipulation. Object shape, texture, grasp points, and continuous spatial relationships are difficult to encode in text, and tasks requiring such information may exceed the capabilities of the current representation. Although the authors acknowledge hybrid memories as future work, the paper does not explore mechanisms to integrate visual or geometric cues, limiting applicability to more physically detailed settings.

5. DDAFT lacks formal justification and comparison to standard RL fine-tuning techniques: While DDAFT intuitively prioritizes difficult trajectories, the method is introduced without strong grounding, and its advantages over established RL-from-feedback or policy fine-tuning approaches remain unclear. In particular, comparisons to more standard baselines such as RFT-style updates, GRPO, or PPO-based fine-tuning would help determine whether DDAFT offers a principled improvement or functions primarily as a heuristic curriculum strategy. Without these comparisons, it is difficult to isolate the contribution of the proposed RL component.

6. Limited analysis of failure cases and memory degradation over long horizons: The paper reports aggregate success metrics but offers limited qualitative analysis of failure modes. In particular, it remains unclear how EPM behaves when memory errors accumulate over long sequences (e.g., misplaced objects, overwritten states, hallucinated object persistence). Without studying memory drift, error correction, or mechanisms for uncertainty estimation, it is difficult to assess reliability in extended, multi-room or multi-task settings.

7. Ambiguity in update policies and memory maintenance strategy: EPM continuously rewrites textual memory, but the rules governing updates (e.g., when to delete stale information, how to handle conflicting observations, how to resolve object identity uncertainty) are not fully articulated. For instance, if an object is temporarily occluded or repositioned off-camera, the system may incorrectly maintain outdated state. An explicit treatment of object permanence, confidence tracking, or uncertainty-aware updates would strengthen the method’s robustness claims.

8. Task and domain diversity is limited: Although the results on PARTNR are compelling, the generality of EPM to broader embodied domains such as navigation, deformable object manipulation, tool use, or embodied household tasks in other environments like BEHAVIOR-1K or Habitat 2.0 is not established. Without cross-domain evaluation, the system’s scalability across diverse embodied tasks remains open.

**Questions:**

1. How do you plan to evaluate the complete EPM planning pipeline in real-world robotic settings, and what challenges do you anticipate in transferring beyond simulation?

2. Have you considered fine-tuning EPM on large-scale real-world embodied datasets (e.g., Ego4D, RT-1/RT-2, BEHAVIOR-1K) to improve generalization for real objects not present in simulation?

3. How does EPM handle conflicting observations, object occlusions, or reappearances? Is there an object permanence or confidence mechanism to avoid stale or incorrect memory states?

4. Can you provide memory drift or long-horizon error analyses? How does the textual memory behave when small memory errors compound over long sequences?

5. What motivates DDAFT over more established RL fine-tuning methods, and have you compared against RFT, PPO-based finetuning, or GRPO?

6. Could you report computational metrics such as update latency, memory insertion cost, and query time, to substantiate the claimed scalability benefits?

7. Why were recent concept-centric memory systems (e.g., ConceptFusion, ConceptGraph) not included as baselines, and how do you expect EPM to compare conceptually and empirically?

8. Do you anticipate difficulty scaling EPM to domains requiring fine-grained perceptual grounding (e.g., grasp point reasoning, deformable objects, tool use), and how might hybrid visual-textual memory help?

9. Have you explored mechanisms to reset, compress, or prune memory to avoid uncontrolled growth or hallucinated persistence in long-horizon tasks?

---

> ### Author Response · Authors · 2025-11-17
>
> We thank the reviewer for the insightful comments. We are glad the reviewer valued the novelty of our memory mechanism, reducing reliance on symbolic querying and multi-module pipelines, and the evaluation in challenging simulation environments with signs of transfer. We address the reviewer’s concerns below, and have updated the main paper and supplementary to reflect the relevant changes.
>
> **It would be valuable to explore whether fine-tuning EPM on diverse real-world embodied dataset.**
>
> Our proposed approach can be trained with different sources of data. To demonstrate the viability of our EPM in open-world settings, we develop a perception-only benchmark with egocentric human videos (Epic-HD) for open-vocabulary object detection and tracking through agent interaction. By training the EPM with data derived from EpicKitchens, the EPM improves detection precision from 0.03 to 0.21 and recall from 0.07 to 0.26. These significant gains from early experimentation indicate strong generalization ability from EPM, pointing to egocentric videos + EPM as a promising avenue toward robot-ready perception systems.
>
>
> **Validation of the planning pipeline in real-world settings.**
>
> We validated our pipeline in real-world settings and reported the results in the original Appendix A.10. Specifically, we deploy the PP-DDAFT planner on a BD Spot robot across four indoor scenarios requiring object localization and manipulation. Each scenario is tested five times with varying clutter configurations, resulting in 20 runs. Our method predicts the correct plan in 70% of tasks (14/20). We updated the main paper to reference these results more clearly.
>
> **No clear advantage compared to DynaMem.**
>
> While DynaMem obtains better perception results than our method in the real world, we respectfully disagree that our method does not show clear advantage over DynaMem. An effective environment representation should not only be accurate but also be useful for planning. As we show in Table 3.1, even when assuming DynaMem has access to ground-truth perception, our approach outperforms DynaMem. Our results show that accurate perception is not enough for effective planning, and that our proposed model enables planners to better use visual information.
>
>
> **Why were recent concept-centric memory systems (e.g., ConceptFusion, ConceptGraph) not included as baselines, and how do you expect EPM to compare conceptually and empirically?**
>
> ConceptGraph and ConceptFusion cannot be applied to our setting, since they assume that the environment is static, whereas in our setting objects can change position or state by the agent. Dynamem is very similar to ConceptGraph, in that they use 2d image features back projected into a point-cloud but adapt the method to handle scene changes. Therefore we chose DynaMem as a representative method for that class of system
>
> **Lacking computational Efficiency Metrics.**
>
> We reported computational efficiency metrics in the original submission's Appendix A.8. EPM takes on average 0.44 seconds to add, remove and update information in the cache, whereas DynaMem takes 1.57 per update. This gap significantly increases when considering the time to complete a full episode, with our approach taking on average 5 minutes, vs. DynaMem takes up to 6 hours. The difference between the two methods arises from the fact that 1) DynaMem requires querying an exhaustive list of objects to build a representation of the scene rather, whereas EPM builds a representation based on the current observation and 2) Our approach takes half the number of steps to complete a task, compared to DynaMem’s iterative approach. The EPM is also more efficient in terms of memory. In our experiments, the EPM’s memory contains at most 3500 tokens, which are approximately 14KB. In contrast, DynaMem stores the full pointcloud in the scene, which is decimated and deduplicated, resulting in 4GB of storage per episode on average. These results show that our approach is not only more accurate but also significantly more efficient than existing methods.

---

> ### Author Response · Authors · 2025-11-17
>
> **Limitations of a text-only memory representation. Do you anticipate difficulty scaling EPM to domains requiring fine-grained perceptual grounding, and how might hybrid visual-textual memory help?**
>
> Our text-only memory provides a compact abstraction of the environment suitable for high-level, long-horizon planning from egocentric observations. The EPM training is general enough to incorporate visual information as continuous features as well. However, the training data should include that information. In general, fine-grained manipulation requires richer and more expressive representations. In those scenarios, a hybrid system could pair the EPM  with specialized visuomotor or VLA modules that operate directly on the current observation and do not require persistent memory. In this hybrid setup, EPM handles long-term spatial reasoning and navigation in large environments, while fine-grained modules are invoked only when the agent reaches the task-relevant context.
>
> **DDAFT lacks formal justification and comparison to standard RL fine-tuning techniques.**
>
> We included additional details of and formalism for DDAFT along with comparisons against other relevant methods in the original submission supplementary material (Appendix A.7), as noted in the end of Sec 4. There we compare against Online-RFT, DAST, and DART-Math as the most relevant baselines. We could not compare against GRPO due to the computational limitations of getting advantage estimates from many different rollouts in the simulator from each step. DDAFT is largely agnostic to the method for advantage estimation (one could use RFT, DPO or GRPO updates). Instead DDAFT differentiates itself from baselines in how the curriculum of difficult episodes is calculated. The most relevant baseline here is DAST [2]. We find DDAFT compares favorably to DAST in terms of sample efficiency (see supplementary Figure 7).
>
>
> **Limited analysis of failure cases and memory degradation over long horizons. Can you provide memory drift or long-horizon error analyses? How does the textual memory behave when small memory errors compound over long sequences?**
>
> To investigate this we present the results of React style planners running with EPM memory and ground truth perception broken down by episode length.
>
> | Method      | Perception | Short (<13 skills) | Medium (13–25 skills) | Long (≥26 skills) |
> |-------------|------------|---------------------|-------------------------|--------------------|
> | HD          | GT         | 0.82                | 0.74                    | 0.26               |
> | HD+DDAFT    | GT         | 0.84                | 0.82                    | 0.31               |
> | HD          | EPM        | 0.82                | 0.50                    | 0.19               |
> | HD+DDAFT    | EPM        | 0.82                | 0.64                    | 0.25
>
> We can see that for all methods, longer episodes are more challenging in general. However the gap in performance between perception EPM and GT is largely due to medium and long episode performance, indicating that compounding perception errors remain an issue. We find that training with DDAFT improves performance in these more challenging episodes by teaching the planner to be robust to perception errors. We included a qualitative example of this planning behavior in the original submission supplementary section A.9.
>
> **Failure analysis in real-world experiments.**
>
> We also analysis for the experiments done in the real-world in Sec A.10. Excluding low-level skill failure from our analysis, we see most failures stem due to failure in perception, which can be categorized as:
> 1. EPM does not detect the active object
> 2. Detects active object as something else
> 3. Adds nonexistent clutter
> 4. Edits the wrong object as being “picked”
>
> These failures then cause downstream planning errors. We see 1+2+4 failures in 5 of 20 runs leading to objective task failure. All of these runs were for tasks with more than one active object. We see failure type 3 in almost all runs, but the trained planner handles additional clutter well as long as failures of type 1,2, or 4 are not present.
>
> Failures of type {1, 2, 4} lead to two kinds of state badness: (a) absence of active object from memory, or (b) active object still in the same place with new object added as “picked”. Interestingly, in case (a) the planner ignores the exact object in the task specification, proceeding to pick any object on the specified receptacle. In case (b) we see the planner keep retrying to pick the actual object (which was already picked, making this a fruitless action) and not make progress on the rest of the plan. We also observed that the planner does not call “explore” in the real-world when right objects are not found at the expected receptacle. Altogether these failures hint at further study into evolving perception and planning together, failure recovery for planning and training/data methodologies that encourage exploration.

---

> > ### Author Response · Authors · 2025-11-17
> >
> > **Ambiguity in update policies and memory maintenance strategy. How does EPM handle conflicting observations, object occlusions, or reappearances?**
> >
> > The EPM maintains a memory via the forward pass of a VLM, which outputs one of the operations described in Sec 3.2 to add, remove or update objects in the memory. To aid in this process, EPM is provided the distance from which the object was previously observed as a proxy for confidence (note that confidence calibration in VLMs is still an open problem [1,2]). EPM learns occlusions and reappearances implicitly through the training data (previously-observed objects do not necessitate Add actions). This approach avoids domain-specific hyperparameter tuning associated with classical long-horizon re-association methods. However, these rules are built-in in the training dataset, which is constructed to handle updating stale information or delete conflicting observations. The mechanism for constructing the dataset is described in detail in A.2.3.
> >
> > **Task and domain diversity is limited.**
> >
> > PARTNR is a comprehensive benchmark for long-horizon planning, containing 100,000 tasks, 200 scenes, and 18,000 object instances. As such, it provides a substantially more challenging testbed for these approaches. For example, the Habitat 2.0 tasks referenced in the review are considerably simpler than PARTNR’s both in task complexity and scene diversity. Deformable object manipulation and tool use are interesting scenarios for future work. Our focus in this work is mainly long-horizon task planning.
> >
> > **Compressing the memory in EPM. Have you explored mechanisms to reset, compress, or prune memory to avoid uncontrolled growth or hallucinated persistence in long-horizon tasks?**
> >
> > This is a very good point. We did find that context growth was an issue for the planner on long episodes. To reduce context growth we only add changes to the EPM into the planner context (see supplementary section A.9 for an example). If the entire entity list from the object memory is added instead, success rates drop significantly (i.e SR falls from 0.46 to 0.32 for the PARTNR planner with EPM memory). Using only the changes to EPM in the planner context causes the planner context to grow by only 32.8 tokens per skill call on average, meaning even episodes which run until timing out (50 skill calls) produce planner contexts less than 3500 tokens. This is easily within the training context length for modern models. We clarified this in the updated Appendix A.4.
> >
> > As for hallucinated objects, the “Remove” operation is trained specifically to address this (Details were included in Appendix section A.2.3). We did not attempt more complicated techniques to compress the planner context (summarization models etc.) as we leave this to future work.
> >
> >
> >
> >
> > *[1] Xuan, W., Zeng, Q., Qi, H., Wang, J., & Yokoya, N. (2025). Seeing is believing, but how much? A comprehensive analysis of verbalized calibration in vision-language models. In Proceedings of the 2025 Conference on Empirical Methods in Natural Language Processing (pp. 1408–1450). Association for Computational Linguistics. Suzhou, China.*
> >
> > *[2] Xue, B., Zhu, Q., Wang, H., Wang, R., Wang, S., Xu, H., ... & Wong, K. F. (2025). Dast: Difficulty-aware self-training on large language models. arXiv preprint arXiv:2503.09029.*

---

### Author Response · Authors · 2025-11-17

We thank the reviewers and the AC for reviewing our paper and providing valuable feedback. In the rebuttal we have clarified the following main points:

1. We have shown that our approach can generalize to multiple domains, by pointing to the real-world experiment we conducted combining the EPM and planning systems (Sec. A.10), and demonstrated that it is possible to build a training dataset with real-world data (in response to RACax).
2. We clarified advantages of our method compared to DynaMem, as well as other related works, and provided further analysis on the qualitative and quantitative performance of our approach (Sec. A.8).
3. We clarified how to scale our method to domains requiring fine-grained reasoning and manipulation (in response to RACax).

We wanted to note that some of the reviewer’s main concerns were already addressed in the paper’s supplementary materials. We have updated the main paper (highlighted in red) to better point to those results.

---

### Author Response · Authors · 2025-11-30
**Summary after review process change**

In light of the changes to the review process after the OpenReview leak, we would like to briefly summarize the comments from the original reviewers and how they were addressed in the rebuttal.

The major concerns raised by reviewers seemed to be *real-world evaluation* (reviewers ACax, U9Bz and JRZU) and *comparisons against other baselines* (reviewers ACax and JRZU). We highlighted that both of these issues were **already addressed in the supplementary** of the original submission, where we included a real-world experiment on the Boston Dynamics spot robot (Supplementary section A.10), and comparisons against SOTA methods for LLM-based planner training (Supplementary section A.7.1.) We have already updated the rebuttal revision of the main paper accordingly so they will not be missed by future readers.

Other concerns, such as error analysis (reviewer ACax) or comparison with VLM-based planners (reviewer JRZU), were addressed directly by providing additional analysis (i.e. long horizon performance breakdown in response to ACax) or references to existing experiments in other work (in response to JRZU). Finally, all minor or clarifying questions were answered directly in the responses to the reviewers.

Given the initially positive ratings of the reviewers, and the fact that all issues were addressed, we believe that if the reviewers were able to update their ratings they would be net-positive.

---

### Meta-Review · Area_Chair_8Mb3 · 2026-01-05

**Summary:**

**Summary:** This paper focuses on embodied agents for long-horizon tasks. It proposes Embodied Perception Memory (EPM) to aid in embodied planning: a VLM that outputs and updates a textual environment representation, making it easy to be integrated with LLM-based planners. It then introduces training/fine-tuning approaches for an LLM-based planner with the EPM attached, based on imitation learning and difficulty-aware RL.

**Strengths (by reviewers).** Well-motivated and well-written paper; the targeted problem is important; a novel, integrative/unified memory module via VLM; EPM leads to seamless integration with LLM-based planners; overall framework is quite complete, lightweight, and can be easily maintained; bypassing explicit querying; two experimental scenarios (perception/planning) that isolate the improvements; notable empirical improvements.

**Weaknesses (by reviewers).** Real-world perception results do not clearly outperform DynaMem; no real-world results on planning; incomplete comparison to existing approaches; lack of computational analysis; limitations of a text-only memory representation; DDAFT lacks formal justification; limited analysis of failure cases; unclear memory update mechanisms; limited task/domain diversity and datasets in experiments; limited novelty: proposed ideas have been explored; missing PARTNR test results; unclear training scalability as it needs privilege information; is dynamic memory really needed?; complex situations and instance associations were not clearly discussed.

**Decision.** The paper received an original score of 5.0 (4, 4, 6, 6). Specifically, reviewers esNZ & U9Bz gave 6; reviewers JRZU & ACax gave 4. The authors provided detailed and easy-to-digest rebuttals, and many questions/concerns have already been addressed in the supplementary materials. Overall, the AC thinks that many concerns/questions were adequately addressed in the rebuttals and revised paper, even though none of the reviewers were involved in the discussion phase. Further, the AC thinks that the paper has clear strengths and it proposed a simple, principled, and extensible framework that could be expanded or built upon by (many) future works. As such, the AC recommends acceptance.

**AC's further suggestion.** The AC has some additional suggestions, partially built upon reviewers' comments.
- Limitation of a text-based, query-free memory. As reviewer ACax also mentioned, a text-only memory representation has limitations. Specifically, *A picture is worth a thousand words.* Namely, representing the dynamic environment by limited texts (several entities with short descriptions) risks omitting important information for specific tasks. The AC likes the honest response by the authors. The AC suggests that the authors incorporate the rebuttal into the camera-ready version and further discuss other limitations or practical considerations: for example, how to ensure the descriptions are comprehensive enough. If EPM has some ad-hoc, rule-based designs/components that need to be modified when applied to different scenarios/datasets, please clarify. This won't hurt the paper, but it would help readers/users understand how to really apply EPM to their problems.
- Regarding the rebuttal to reviewer U9Bz's concerns on novelty, especially [1], the AC suggests the authors dive deeper. Specifically, a multi-stage/module approach was a standard way to perceive the environment before the presence of integrated VLMs, and even with VLMs, standalone object detections and segmentation models can still outperform in certain (complex) environments.
- Following the above comment, the AC suggests that the authors adjust the claims/tones in the introduction. The current version reads a bit over-claiming.
- Regarding the rebuttal to reviewer ACax's concerns, the AC finds that the details of EpicKitchens are missing; there is no comparison of real-world planning experiments (i.e., no baselines) despite the results of the proposed method.
- The authors are encouraged to have better references (or summaries) of the supplementary materials in the main paper.

**Reviewer Concerns:**

**Reviewer ACax.**
- **Addressed:** Comparison to DynaMem; real-world planning results; incomplete comparisons; lack of computational analysis; limitations of a text-only memory representation; DDAFT's justifications; limited analysis of failure cases; unclear memory update mechanisms; limited task/domain diversity and datasets in experiments.
- **Remaining:** fine-tuning EPM on diverse real-world embodied dataset (more details needed)

**Reviewer U9Bz.**
- **Addressed:** Limited novelty; missing PARTNR test results; unclear training scalability, as it needs privilege information (AC's comment: it becomes indeed clear that preparing training data well is key to advance AI); multi-object cases
- **Remaining:** None

**Reviewer JRZU.**
- **Addressed:** Why not directly use VLMs? Why is a memory needed? VLM vs. LLM for planning.
- **Remaining:** Additional benchmarks (the authors did not provide more, but provided an argument); the need for dynamic memories.

**Reviewer esNZ.**
- **Addressed:** clarification questions
- **Remaining:** None

**Reviewer Scores:**

**Reviewer ACax (4 to 6):** The reviewer has extensive questions/concerns, and the AC thinks the rebuttal has addressed most of them.

**Reviewer U9Bz (6 to 6):** Even though the rebuttal has addressed reviewers' concerns, the AC does not think it would increase the score to 8.

**Reviewer JRZU (4 to 5):** The authors responded to all the reviewers' questions/concerns, but some may not be fully addressed. Specifically, the AC thinks the authors' arguments on using PARTNR and not using other well-known datasets are valid, but the reviewer might have different opinions. Thus, the AC thinks the score might be increased from 4 to 5 (not 6).

**Reviewer esNZ (6 to 6).** Even though the rebuttal has addressed reviewers' concerns, the AC does not think it would increase the score to 8.

---

### Decision · Program_Chairs · 2026-01-26

Accept (Poster)